# PER-TENSOR FIXED-POINT QUANTIZATION OF THE BACK-PROPAGATION ALGORITHM

**Charbel Sakr & Naresh Shanbhag**
Department of Electrical and Computer Engineering
University of Illinois at Urbana-Champaign
Illinois, IL 61801, USA
`{sakr2,shanbhag}@illinois.edu`

## ABSTRACT

The high computational and parameter complexity of neural networks makes their training very slow and difficult to deploy on energy and storage-constrained computing systems. Many network complexity reduction techniques have been proposed including fixed-point implementation. However, a systematic approach for designing full fixed-point training and inference of deep neural networks remains elusive. We describe a precision assignment methodology for neural network training in which all network parameters, i.e., activations and weights in the feedforward path, gradients and weight accumulators in the feedback path, are assigned close to minimal precision. The precision assignment is derived analytically and enables tracking the convergence behavior of the full precision training, known to converge a priori. Thus, our work leads to a systematic methodology of determining suitable precision for fixed-point training. The near optimality (minimality) of the resulting precision assignment is validated empirically for four networks on the CIFAR-10, CIFAR-100, and SVHN datasets. The complexity reduction arising from our approach is compared with other fixed-point neural network designs.

## 1 INTRODUCTION

Though deep neural networks (DNNs) have established themselves as powerful predictive models achieving human-level accuracy on many machine learning tasks (He et al., 2016), their excellent performance has been achieved at the expense of a very high *computational* and *parameter* complexity. For instance, AlexNet (Krizhevsky et al., 2012) requires over $800 \times 10^6$ multiply-accumulates (MACs) per image and has 60 million parameters, while Deepface (Taigman et al., 2014) requires over $500 \times 10^6$ MACs/image and involves more than 120 million parameters. DNNs' enormous computational and parameter complexity leads to high energy consumption (Chen et al., 2017), makes their training via the *stochastic gradient descent* (SGD) algorithm very slow often requiring hours and days (Goyal et al., 2017), and inhibits their deployment on energy and resource-constrained platforms such as mobile devices and autonomous agents.

A fundamental problem contributing to the high computational and parameter complexity of DNNs is their realization using 32-b floating-point (FL) arithmetic in GPUs and CPUs. Reduced-precision representations such as *quantized FL* (QFL) and *fixed-point* (FX) have been employed in various combinations to both training and inference. Many employ FX during inference but train in FL, e.g., fully binarized neural networks (Hubara et al., 2016) use 1-b FX in the forward inference path but the network is trained in 32-b FL. Similarly, Gupta et al. (2015) employs 16-b FX for all tensors except for the internal accumulators which use 32-b FL, and 3-level QFL gradients were employed (Wen et al., 2017; Alistarh et al., 2017) to accelerate training in a distributed setting. Note that while QFL reduces storage and communication costs, it does not reduce the computational complexity as the arithmetic remains in 32-b FL.

Thus, none of the previous works address the fundamental problem of realizing *true fixed-point DNN training*, i.e., an SGD algorithm in which all parameters/variables and all computations are implemented in FX with *minimum precision* required to *guarantee the network's inference/prediction accuracy* and *training convergence*. The reasons for this gap are numerous including: 1) quantization

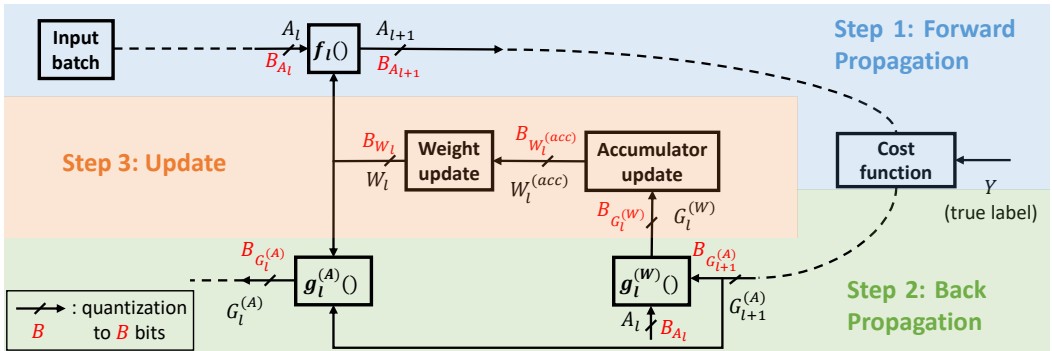

Figure 1: Problem setup: FX training at layer $l$ of a DNN showing the quantized tensors and the associated precision configuration $C_l = (B_{W_l}, B_{A_l}, B_{G_l^{(W)}}, B_{G_{l+1}^{(A)}}, B_{W_l^{(acc)}})$.

errors propagate to the network output thereby directly affecting its accuracy (Lin et al., 2016); 2) precision requirements of different variables in a network are interdependent and involve hard-to-quantify trade-offs (Sakr et al., 2017); 3) proper quantization requires the knowledge of the dynamic range which may not be available (Pascanu et al., 2013); and 4) quantization errors may accumulate during training and can lead to stability issues (Gupta et al., 2015).

Our work makes a major advance in closing this gap by proposing a systematic methodology to obtain *close-to-minimum* per-layer precision requirements of an FX network that guarantees statistical similarity with full precision training. In particular, we jointly address the challenges of quantization noise, inter-layer and intra-layer precision trade-offs, dynamic range, and stability. As in (Sakr et al., 2017), we do assume that a fully-trained baseline FL network exists and one can observe its learning behavior. While, in principle, such assumption requires extra FL computation prior to FX training, it is to be noted that much of training is done in FL anyway. For instance, FL training is used in order to establish benchmarking baselines such as AlexNet (Krizhevsky et al., 2012), VGG-Net (Simonyan and Zisserman, 2014), and ResNet (He et al., 2016), to name a few. Even if that is not the case, in practice, this assumption can be accounted for via a warm-up FL training on a small held-out portion of the dataset (Dwork et al., 2015).

Applying our methodology to three benchmarks reveals several lessons. First and foremost, our work shows that it is possible to FX quantize all variables including back-propagated gradients even though their dynamic range is unknown (Köster et al., 2017). Second, we find that the per-layer weight precision requirements decrease from the input to the output while those of the activation gradients and weight accumulators increase. Furthermore, the precision requirements for residual networks are found to be uniform across layers. Finally, hyper-precision reduction techniques such as weight and activation binarization (Hubara et al., 2016) or gradient ternarization (Wen et al., 2017) are not as efficient as our methodology since these do not address the fundamental problem of realizing true fixed-point DNN training.

We demonstrate FX training on three deep learning benchmarks (CIFAR-10, CIFAR-100, SVHN) achieving *high fidelity* to our FL baseline in that we observe no loss of accuracy higher then **0.56%** in all of our experiments. Our precision assignment is further shown to be *within 1-b per-tensor* of the minimum. We show that our precision assignment methodology reduces representational, computational, and communication costs of training by up to **6×**, **8×**, and **4×**, respectively, compared to the FL baseline and related works.

## 2 PROBLEM SETUP, NOTATION, AND METRICS

We consider a $L$-layer DNN deployed on a $M$-class classification task using the setup in Figure 1. We denote the *precision configuration* as the $L \times 5$ matrix $C = (B_{W_l}, B_{A_l}, B_{G_l^{(W)}}, B_{G_{l+1}^{(A)}}, B_{W_l^{(acc)}})_{l=1}^{L}$ whose $l^{\text{th}}$ row consists of the precision (in bits) of weight $W_l$ ($B_{W_l}$), activation $A_l$ ($B_{A_l}$), weight

gradient $G_l^{(W)}$ ($B_{G_l^{(W)}}$), activation gradient $G_{l+1}^{(A)}$ ($B_{G_{l+1}^{(A)}}$), and internal weight accumulator $W_l^{(acc)}$ ($B_{W_l^{(acc)}}$) tensors at layer $l$. This DNN quantization setup is summarized in Appendix A.

## 2.1 FIXED-POINT CONSTRAINTS & DEFINITIONS

We present definitions/constraints related to fixed-point arithmetic based on the design of fixed-point adaptive filters and signal processing systems (Parhi, 2007):

- A *signed* fixed-point scalar $a$ with precision $B_A$ and binary representation $R_A = (a_0, a_1, \ldots, a_{B_A-1}) \in \{0,1\}^{B_A}$ is equal to: $a = r_A \left( -a_0 + \sum_{i=1}^{B_A-1} 2^{-i} a_i \right)$, where $r_A$ is the predetermined dynamic range (PDR) of $a$. The PDR is constrained to be a **constant power of 2** to minimize hardware overhead.
- An *unsigned* fixed-point scalar $a$ with precision $B_A$ and binary representation $R_A = (a_0, a_1, \ldots, a_{B_A-1}) \in \{0,1\}^{B_A}$ is equal to: $a = r_A \sum_{i=0}^{B_A-1} 2^{-i} a_i$.
- A fixed-point scalar $a$ is called *normalized* if $r_A = 1$.
- The precision $B_A$ is determined as: $B_A = \log_2 \frac{r_A}{\Delta_A} + 1$, where $\Delta_A$ is the *quantization step size* which is the value of the *least significant bit (LSB)*.
- An *additive model* for quantization is assumed: $a = \tilde{a} + q_a$, where $a$ is the fixed-point number obtained by quantizing the floating-point scalar $\tilde{a}$, $q_a$ is a random variable uniformly distributed on the interval $\left[ -\frac{\Delta_A}{2}, \frac{\Delta_A}{2} \right]$, and the quantization noise variance is $Var(q_a) = \frac{\Delta_A^2}{12}$. The notion of quantization noise is most useful when there is limited knowledge of the distribution of $\tilde{a}$.
- The *relative quantization bias* $\eta_A$ is the offset: $\eta_A = \frac{|\Delta_A - \mu_A|}{\mu_A}$, where the first unbiased quantization level $\mu_A = \mathbb{E}\left[ \tilde{a} \middle| \tilde{a} \in I_1 \right]$ and $I_1 = \left[ \frac{\Delta_A}{2}, \frac{3\Delta_A}{2} \right]$. The notion of quantization bias is useful when there is some knowledge of the distribution of $\tilde{a}$.
- The *reflected quantization noise variance* from a tensor $T$ to a scalar $\alpha = f(T)$, for an arbitrary function $f()$, is : $V_{T \to \alpha} = E_{T \to \alpha} \frac{\Delta_T^2}{12}$, where $\Delta_T$ is the quantization step of $T$ and $E_{T \to \alpha}$ is the *quantization noise gain* from $T$ to $\alpha$.
- The *clipping rate* $\beta_T$ of a tensor $T$ is the probability: $\beta_T = \Pr\left( \{|t| \geq r_T : t \in T\} \right)$, where $r_T$ is the PDR of $T$.

## 2.2 COMPLEXITY METRICS

We use a set of metrics inspired by those introduced by Sakr et al. (2017) which have also been used by Wu et al. (2018a). These metrics are algorithmic in nature which makes them easily reproducible.

- *Representational Cost* for weights ($\mathcal{C}_W$) and activations ($\mathcal{C}_A$):
  $\mathcal{C}_W = \sum_{l=1}^L |W_l| \left( B_{W_l} + B_{G_l^{(W)}} + B_{W_l^{(acc)}} \right)$ & $\mathcal{C}_A = \sum_{l=1}^L |A_l| \left( B_{A_l} + B_{G_{l+1}^{(A)}} \right)$,
  which equals the total number of bits needed to represent the weights, weight gradients, and internal weight accumulators ($\mathcal{C}_W$), and those for activations and activation gradients ($\mathcal{C}_A$). [1]
- *Computational Cost* of training: $\mathcal{C}_M = \sum_{l=1}^L |A_{l+1}| D_l \left( B_{W_l} B_{A_l} + B_{W_l} B_{G_{l+1}^{(A)}} + B_{A_l} B_{G_{l+1}^{(A)}} \right)$,
  where $D_l$ is the dimensionality of the dot product needed to compute one output activation at layer $l$. This cost is a measure of the number of 1-b full adders (FAs) utilized for all multiplications in one back-prop iteration. [2]
- *Communication Cost*: $\mathcal{C}_C = \sum_{l=1}^L |W_l| B_{G_l^{(W)}}$, which represents cost of communicating weight gradients in a distributed setting (Wen et al., 2017; Alistarh et al., 2017).

---

[1] We use the notation $|T|$ to denote the number of elements in tensor $T$. Unquantized tensors are assumed to have a 32-b FL representation, which is the single-precision in a GPU.

[2] When considering 32-b FL multiplications, we ignore the cost of exponent addition thereby favoring the FL (conventional) implementation. Boundary effects (in convolutions) are neglected.

## 3  PRECISION ASSIGNMENT METHODOLOGY AND ANALYSIS

We aim to obtain a minimal or close-to-minimal precision configuration $C_o$ of a FX network such that the mismatch probability $p_m = \Pr\{\hat{Y}_{fl} \neq \hat{Y}_{fx}\}$ between its predicted label ($\hat{Y}_{fx}$) and that of an associated FL network ($\hat{Y}_{fl}$) is bounded, and the convergence behavior of the two networks is similar.

Hence, we require that: (1) all quantization noise sources in the forward path contribute identically to the mismatch budget $p_m$ (Sakr et al., 2017), (2) the gradients be properly clipped in order to limit the dynamic range (Pascanu et al., 2013), (3) the accumulation of quantization noise bias in the weight updates be limited (Gupta et al., 2015), (4) the quantization noise in activation gradients be limited as these are back-propagated to calculate the weight gradients, and (5) the precision of weight accumulators should be set so as to avoid premature stoppage of convergence (Goel and Shanbhag, 1998). The above insights can be formally described via the following five quantization criteria.

**Criterion 1.** *Equalizing Feedforward Quantization Noise (EFQN) Criterion.* The reflected quantization noise variances onto the mismatch probability $p_m$ from all feedforward weights ($\{V_{W_l \to p_m}\}_{l=1}^L$) and activations ($\{V_{A_l \to p_m}\}_{l=1}^L$) should be equal:

$$V_{W_1 \to p_m} = \ldots = V_{W_L \to p_m} = V_{A_1 \to p_m} = \ldots = V_{A_L \to p_m}$$

**Criterion 2.** *Gradient Clipping (GC) Criterion.* The clipping rates of weight ($\{\beta_{G_l^{(W)}}\}_{l=1}^L$) and activation ($\{\beta_{G_{l+1}^{(A)}}\}_{l=1}^L$) gradients should be less than a maximum value $\beta_0$:

$$\beta_{G_l^{(W)}} < \beta_0 \quad \& \quad \beta_{G_{l+1}^{(A)}} < \beta_0 \quad \text{for } l = 1 \ldots L.$$

**Criterion 3.** *Relative Quantization Bias (RQB) Criterion.* The relative quantization bias of weight gradients ($\{\eta_{G_l^{(W)}}\}_{l=1}^L$) should be less than a maximum value $\eta_0$:

$$\eta_{G_l^{(W)}} < \eta_0 \quad \text{for } l = 1 \ldots L.$$

**Criterion 4.** *Back-propagated Quantization Noise (BQN) Criterion.* The reflected quantization noise variance $V_{G_{l+1}^{(A)} \to \Sigma_l}$, i.e., the total sum of element-wise variances of $G_l^{(W)}$ reflected from quantizing $G_{l+1}^{(A)}$, should be less than $V_{G_l^{(W)} \to \Sigma_l}$:

$$V_{G_{l+1}^{(A)} \to \Sigma_l} \leq V_{G_l^{(W)} \to \Sigma_l} \quad \text{for } l = 1 \ldots L.$$

where $\Sigma_l$ is the total sum of element-wise variances of $G_l^{(W)}$.

**Criterion 5.** *Accumulator Stopping (AS) Criterion.* The quantization noise of the internal accumulator should be zero, equivalently:

$$V_{W_l^{(acc)} \to \Sigma_l^{(acc)}} = 0 \quad \text{for } l = 1 \ldots L$$

where $V_{W_l^{(acc)} \to \Sigma_l^{(acc)}}$ is the reflected quantization noise variance from $W_l^{(acc)}$ to $\Sigma_l^{(acc)}$, its total sum of element-wise variances.

Further explanations and motivations behind the above criteria are presented in Appendix B. The following claim ensures the satisfiability of the above criteria. This leads to closed form expressions for the precision requirements we are seeking and completes our methodology. The validity of the claim is proved in Appendix C.

**Claim 1.** *Satisfiability of Quantization Criteria.* The five quantization criteria (EFQN, GC, RQB, BQN, AS) are satisfied if:

• The precisions $B_{W_l}$ and $B_{A_l}$ are set as follows:

$$B_{W_l} = rnd\left(\log_2\left(\sqrt{\frac{E_{W_l \to p_m}}{E^{(\min)}}}\right)\right) + B^{(\min)} \quad \& \quad B_{A_l} = rnd\left(\log_2\left(\sqrt{\frac{E_{A_l \to p_m}}{E^{(\min)}}}\right)\right) + B^{(\min)}$$

$$(1)$$

for $l = 1 \ldots L$, where $rnd()$ denotes the rounding operation, $E_{W_l \to p_m}$ and $E_{A_l \to p_m}$ are the weight and activation quantization noise gains at layer $l$, respectively, $B^{(\min)}$ is a reference minimum precision, and $E^{(\min)} = \min\left(\{E_{W_l \to p_m}\}_{l=1}^L, \{E_{A_l \to p_m}\}_{l=1}^L\right)$.

- The weight and activation gradients PDRs are lower bounded as follows:

$$r_{G_l^{(W)}} \geq 2\sigma_{G_l^{(W)}}^{(\text{max})} \quad \& \quad r_{G_{l+1}^{(A)}} \geq 4\sigma_{G_{l+1}^{(A)}}^{(\text{max})} \quad \text{for } l = 1 \ldots L \tag{2}$$

where $\sigma_{G_l^{(W)}}^{(\text{max})}$ and $\sigma_{G_{l+1}^{(A)}}^{(\text{max})}$ are the largest recorded estimates of the weight and activation gradients standard deviations $\sigma_{G_l^{(W)}}$ and $\sigma_{G_{l+1}^{(A)}}$, respectively.

- The weight and activation gradients quantization step sizes are upper bounded as follows:

$$\Delta_{G_l^{(W)}} < \frac{\sigma_{G_l^{(W)}}^{(\text{min})}}{4} \quad \& \quad \Delta_{G_{l+1}^{(A)}} < \frac{\Delta_{G_l^{(W)}}}{\sqrt{\lambda_{G_{l+1}^{(A)} \to G_l^{(W)}}^{(\text{max})}}} \left( \frac{\left| G_l^{(W)} \right|}{\left| G_{l+1}^{(A)} \right|} \right)^{1/4} \quad \text{for } l = 1 \ldots L \tag{3}$$

where $\sigma_{G_l^{(W)}}^{(\text{min})}$ is the smallest recorded estimate of $\sigma_{G_l^{(W)}}$ and $\lambda_{G_{l+1}^{(A)} \to G_l^{(W)}}^{(\text{max})}$ is the largest singular value of the square-Jacobian (Jacobian matrix with squared entries) of $G_l^{(W)}$ with respect to $G_{l+1}^{(A)}$.

- The accumulator PDR and step size satisfy:

$$r_{W_l^{(acc)}} \geq 2^{-Bw_l} \quad \& \quad \Delta_{W_l^{(acc)}} < \gamma^{(\text{min})} \Delta_{G_l^{(W)}} \quad \text{for } l = 1 \ldots L \tag{4}$$

where $\gamma^{(\text{min})}$ is the smallest value of the learning rate used during training.

**Practical considerations:** Note that one of the $2L$ feedforward precisions will equal $B^{(\text{min})}$. The formulas to compute the quantization noise gains are given in Appendix C and require only one forward-backward pass on an estimation set. We would like the EFQN criterion to hold upon convergence; hence, (1) is computed using the converged model from the FL baseline. For backward signals, setting the values of PDR and LSB is sufficient to determine the precision using the identity $B_A = \log_2 \frac{r_A}{\Delta_A} + 1$, as explained in Section 2.1. As per Claim 1, estimates of the second order statistics, e.g., $\sigma_{G_l^{(W)}}$ and $\sigma_{G_{l+1}^{(A)}}$, of the gradient tensors, are required. These are obtained via tensor spatial averaging, so that one estimate per tensor is required, and updated in a moving window fashion, as is done for normalization parameters in BatchNorm (Ioffe and Szegedy, 2015). Furthermore, it might seem that computing the Jacobian in (3) is a difficult task; however, the values of its elements are already computed by the back-prop algorithm, requiring no additional computations (see Appendix C). Thus, the Jacobians (at different layers) are also estimated during training. Due to the typical very large size of modern neural networks, we average the Jacobians spatially, i.e., the activations are aggregated across channels and mini-batches while weights are aggregated across filters. This is again inspired by the work on Batch Normalization (Ioffe and Szegedy, 2015) and makes the probed Jacobians much smaller.

## 4 NUMERICAL RESULTS

We conduct numerical simulations in order to illustrate the validity of the predicted precision configuration $C_o$ and investigate its minimality and benefits. We employ three deep learning benchmarking datasets: CIFAR-10, CIFAR-100 (Krizhevsky and Hinton, 2009), and SVHN (Netzer et al., 2011). All experiments were done using a Pascal P100 NVIDIA GPU. We train the following networks:

- CIFAR-10 ConvNet: a 9-layer convolutional neural network trained on the CIFAR-10 dataset described as $2 \times (64C3) - MP2 - 2 \times (128C3) - MP2 - 2 \times (256C3) - 2 \times (512FC) - 10$ where $C3$ denotes $3 \times 3$ convolutions, $MP2$ denotes $2 \times 2$ max pooling operation, and $FC$ denotes fully connected layers.
- SVHN ConvNet: the same network as the CIFAR-10 ConvNet, but trained on the SVHN dataset.
- CIFAR-10 ResNet: a wide deep residual network (Zagoruyko and Komodakis, 2016) with ResNet-20 architecture but having 8 times as many channels per layer compared to (He et al., 2016).
- CIFAR-100 ResNet: same network as CIFAR-10 ResNet save for the last layer to match the number of classes (100) in CIFAR-100.

A step by step description of the application of our method to the above four networks is provided in Appendix E. We hope the inclusion of these steps would: (1) clarify any ambiguity the reader may have from the previous section and (2) facilitate the reproduction of our results.

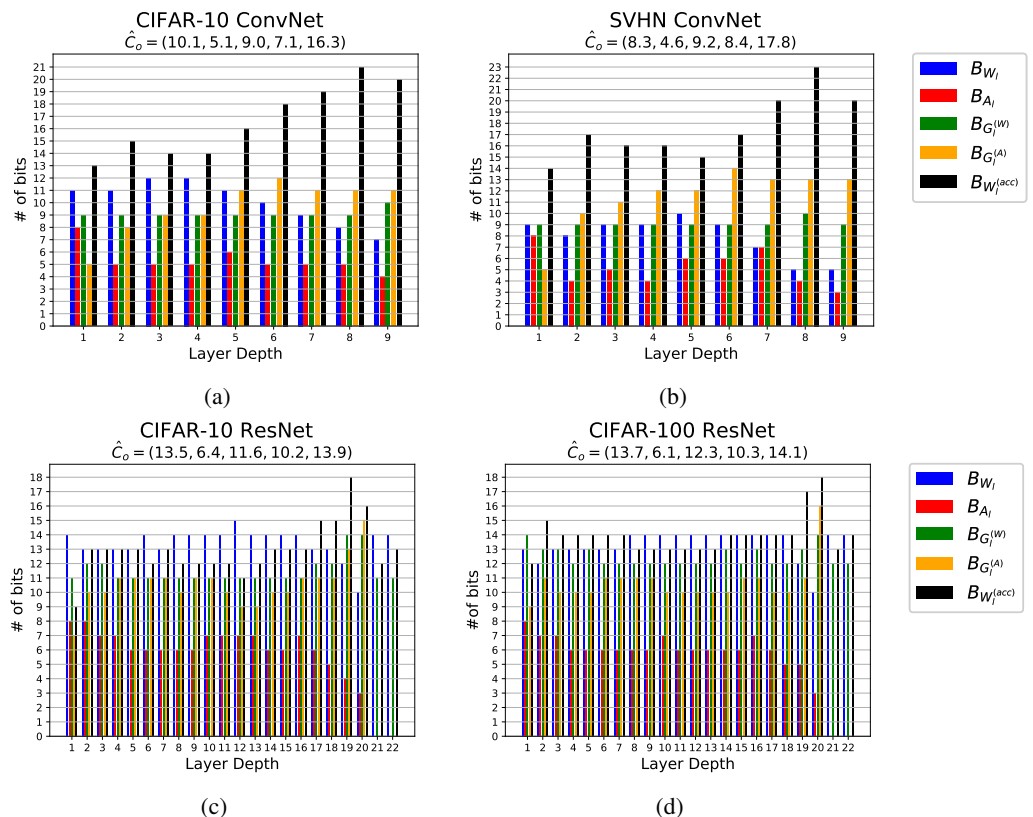

Figure 2: The predicted precision configurations $C_o$ for the CIFAR-10 ConvNet (a), SVHN ConvNet (b), CIFAR-10 ResNet (c), and CIFAR-100 ResNet (d). For each network, the 5-tuple $\hat{C}_o$ represents the average number of bits per tensor type. For the ResNets, layer depths 21 and 22 correspond to the strided convolutions in the shortcut connections of residual blocks 4 and 7, respectively. Activation gradients go from layer 2 to $L + 1$ and are "shifted to the left" in order to be aligned with the other tensors.

### 4.1 PRECISION CONFIGURATION $C_o$ & CONVERGENCE

The precision configuration $C_o$, with target $p_m \leq 1\%$, $\beta_0 \leq 5\%$, and $\eta_0 \leq 1\%$, via our proposed method is depicted in Figure 2 for each of the four networks considered. We observe that $C_o$ is dependent on the network *type*. Indeed, the precisions of the two ConvNets follow similar trends as do those the two ResNets. Furthermore, the following observations are made for the ConvNets:

- weight precision $B_{W_l}$ *decreases as depth increases*. This is consistent with the observation that weight perturbations in the earlier layers are the most destructive (Raghu et al., 2017).
- the precisions of activation gradients ($B_{G_l^{(A)}}$) and internal weight accumulators ($B_{W_l^{(acc)}}$) *increases as depth increases* which we interpret as follows: (1) the back-propagation of gradients is the *dual* of the forward-propagation of activations, and (2) accumulators store the *most information* as their precision is the highest.
- the precisions of the weight gradients ($B_{G_l^{(W)}}$) and activations ($B_{A_l}$) are *relatively constant across layers*.

Interestingly, for ResNets, the precision is mostly uniform across the layers. Furthermore, the gap between $B_{W_l^{(acc)}}$ and the other precisions is not as pronounced as in the case of ConvNets. This suggests that information is spread equally among all signals which we speculate is due to the shortcut connections preventing the *shattering* of information (Balduzzi et al., 2017).

FX training curves in Figure 3 indicate that $C_o$ leads to *convergence* and consistently track FL curves with *close fidelity*. This validates our analysis and justifies the choice of $C_o$.

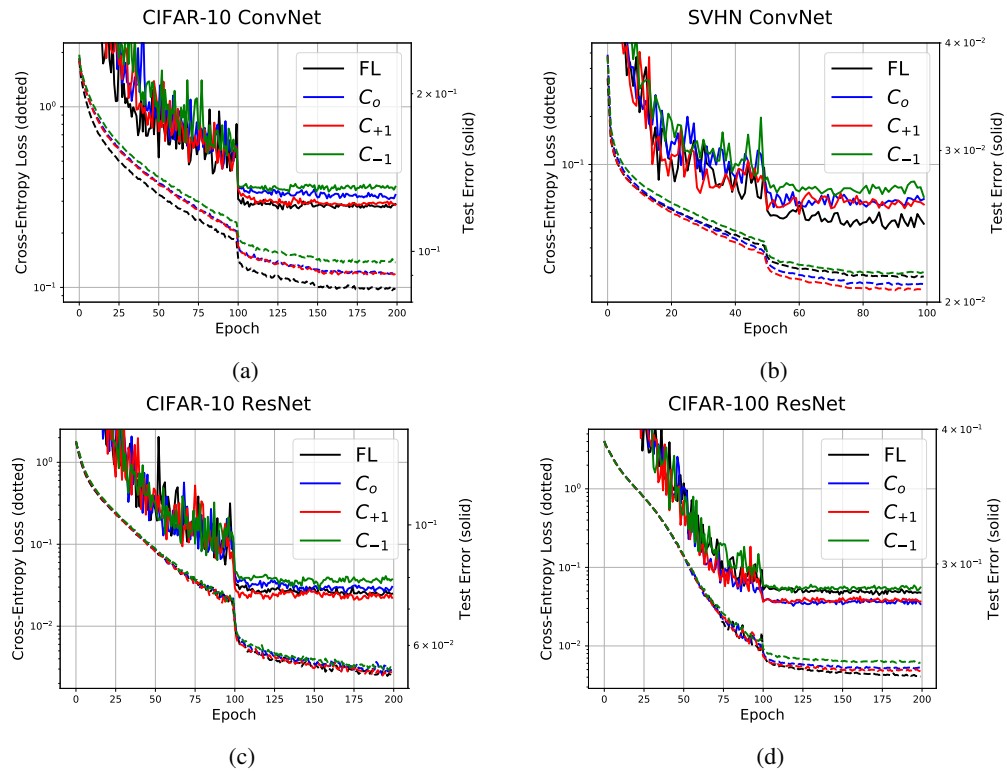

Figure 3: Convergence curves for the CIFAR-10 ConvNet (a), SVHN ConvNet (b), CIFAR-10 ResNet (c), and CIFAR-100 ResNet (d) including FL training as well as FX training with precision configurations $C_o$, $C_1$, and $C_{-1}$.

## 4.2 NEAR MINIMALITY OF $C_o$

To determine that $C_o$ is a close-to-minimal precision assignment, we compare it with: (a) $C_{+1} = C_o + \mathbf{1}_{L \times 5}$, and (b) $C_{-1} = C_o - \mathbf{1}_{L \times 5}$, where $\mathbf{1}_{L \times 5}$ is an $L \times 5$ matrix with each entry equal to $1^3$, i.e., we perturb $C_o$ by 1-b in either direction. Figure 3 also contains the convergence curves for the two new configurations. As shown, $C_{-1}$ always results in a noticeable gap compared to $C_o$ for both the loss function (except for the CIFAR-10 ResNet) and the test error. Furthermore, $C_{+1}$ offers no observable improvements over $C_o$ (except for the test error of CIFAR-10 ConvNet). These results support our contention that $C_o$ is *close-to-minimal* in that increasing the precision above $C_o$ leads to diminishing returns while reducing precision below $C_o$ leads to a noticeable degradation in accuracy. Additional experimental results provided in Appendix D support our contention regarding the near minimality of $C_o$. Furthermore, by studying the impact of quantizing specific tensors we determine that that the accuracy is most sensitive to the precision assigned to weights and activation gradients.

## 4.3 COMPLEXITY VS. ACCURACY

We would like to quantify the reduction in training cost and expense in terms of accuracy resulting from our proposed method and compare them with those of other methods. Importantly, for a fair comparison, the *same* network architecture and training procedure are used. We report $\mathcal{C}_W$, $\mathcal{C}_A$, $\mathcal{C}_M$, $\mathcal{C}_C$, and test error, for each of the four networks considered for the following training methods:

- baseline FL training and FX training using $C_o$,
- binarized network (BN) training, where feedforward weights and activations are binary (constrained to $\pm 1$) while gradients and accumulators are in floating-point and activation gradients are back-

---

[3]PDRs are unchanged across configurations, except for $r_{W_l^{(acc)}}$ as per (4).

Table 1: Complexity ($\mathcal{C}_W$, $\mathcal{C}_A$, $\mathcal{C}_M$, and $\mathcal{C}_C$) and accuracy (test error) for the floating-point (FL), fixed-point (FX) with precision configuration $C_o$, binarized network (BN), stochastic quantization (SQ), and TernGrad (TG) training schemes.

| | $\mathcal{C}_W$ ($10^6$b) | $\mathcal{C}_A$ ($10^6$b) | $\mathcal{C}_M$ ($10^9$FA) | $\mathcal{C}_C$ ($10^6$b) | Test Error | $\mathcal{C}_W$ ($10^6$b) | $\mathcal{C}_A$ ($10^6$b) | $\mathcal{C}_M$ ($10^9$FA) | $\mathcal{C}_C$ ($10^6$b) | Test Error |
|---|---|---|---|---|---|---|---|---|---|---|
| | **CIFAR-10 ConvNet** | | | | | **SVHN ConvNet** | | | | |
| FL | 148 | 9.3 | 94.4 | 49 | 12.02% | 148 | 9.3 | 94.4 | 49 | **2.43**% |
| **FX** ($C_o$) | **56.5** | **1.7** | 11.9 | 14 | 12.58% | **54.3** | **1.9** | 10.5 | 14 | 2.58% |
| BN | 100 | 4.7 | **2.8** | 49 | 18.50% | 100 | 4.7 | **2.8** | 49 | 3.60% |
| SQ | 78.8 | **1.7** | 11.9 | 14 | **11.32**% | 76.3 | **1.9** | 10.5 | 14 | 2.73% |
| TG | 102 | 9.3 | 94.4 | **3.1** | 12.49% | 102 | 9.3 | 94.4 | **3.1** | 3.65% |
| | **CIFAR-10 ResNet** | | | | | **CIFAR-100 ResNet** | | | | |
| FL | 1784 | 96 | 4319 | 596 | 7.42% | 1789 | 97 | 4319 | 597 | 28.06% |
| **FX** ($C_o$) | **726** | **25** | 785 | 216 | 7.51% | **750** | **25** | 776 | 216 | **27.43**% |
| BN | 1208 | 50 | **128** | 596 | **7.24**% | 1211 | 50 | **128** | 597 | 29.35% |
| SQ | 1062 | **25** | 785 | 216 | 7.42% | 1081 | **25** | 776 | 216 | 28.03% |
| TG | 1227 | 96 | 4319 | **37.3** | 7.94% | 1230 | 97 | 4319 | **37.3** | 30.62% |

propagated via the straight through estimator (Bengio et al., 2013) as was done in (Hubara et al., 2016),

- fixed-point training with stochastic quantization (SQ). As was done in (Gupta et al., 2015), we quantize feedforward weights and activations as well as all gradients, but accumulators are kept in floating-point. The precision configuration (excluding accumulators) is inherited from $C_o$ (hence we determine exactly how much stochastic quantization helps),
- training with ternarized gradients (TG) as was done in TernGrad (Wen et al., 2017). All computations are done in floating-point but weight gradients are ternarized according to the instantaneous tensor spatial standard deviations $\{-2.5\sigma, 0, 2.5\sigma\}$ as was suggested by Wen et al. (2017). To compute costs, we assume all weight gradients use two bits although they are not really fixed-point and do require computation of 32-b floating-point scalars for every tensor.

The comparison is presented in Table 1. The first observation is a massive complexity reduction compared to FL. For instance, for the CIFAR-10 ConvNet, the complexity reduction is $2.6\times$ ($= 148/56.5$), $5.5\times$ ($= 9.3/1.7$), $7.9\times$ ($= 94.4/11.9$), and $3.5\times$ ($= 49/14$) for $\mathcal{C}_W$, $\mathcal{C}_A$, $\mathcal{C}_M$, and $\mathcal{C}_C$, respectively. Similar trends are observed for the other four networks. Such complexity reduction comes at the expense of no more than **0.56%** increase in test error. For the CIFAR-100 network, the accuracy when training in fixed-point is even better than that of the baseline.

The representational and communication costs of BN is significantly greater than that of FX because the gradients and accumulators are kept in full precision, which masks the benefits of binarizing feedforward tensors. However, benefits are noticeable when considering the computational cost which is lowest as binarization eliminates multiplications. Furthermore, binarization causes a severe accuracy drop for the ConvNets but surprisingly not for the ResNets. We speculate that this is due to the high dimensional geometry of ResNets (Anderson and Berg, 2017).

As for SQ, since $C_o$ was inherited, all costs are identical to FX, save for $\mathcal{C}_W$ which is larger due to full precision accumulators. Furthermore, SQ has a positive effect only on the CIFAR-10 ConvNet where it clearly acted as a regularizer.

TG does not provide complexity reductions in terms of representational and computational costs which is expected as it only compresses weight gradients. Additionally, the resulting accuracy is slightly worse than that of all other considered schemes, including FX. Naturally, it has the lowest communication cost as weight gradients are quantized to just 2-b.

# 5    DISCUSSION

## 5.1    RELATED WORKS

Many works have addressed the general problem of reduced precision/complexity deep learning.

**Reducing the complexity of inference (forward path):** several research efforts have addressed the problem of realizing a DNN's inference path in FX. For instance, the works in (Lin et al., 2016; Sakr et al., 2017) address the problem of precision assignment. While Lin et al. (2016) proposed a non-uniform precision assignment using the signal-to-quantization-noise ratio (SQNR) metric, Sakr et al. (2017) analytically quantified the trade-off between activation and weight precisions while providing minimal precision requirements of the inference path computations that bounds the probability $p_m$ of a mismatch between predicted labels of the FX and its FL counterpart. An orthogonal approach which can be applied on top of quantization is pruning (Han et al., 2015). While significant inference efficiency can be achieved, this approach incurs a substantial training overhead. A subset of the FX training problem was addressed in binary weighted neural networks (Courbariaux et al., 2015; Rastegari et al., 2016) and fully binarized neural networks (Hubara et al., 2016), where direct training of neural networks with *pre-determined precisions* in the inference path was explored with the feedback path computations being done in 32-b FL.

**Reducing the complexity of training (backward path):** finite-precision training was explored in (Gupta et al., 2015) which employed stochastic quantization in order to counter quantization bias accumulation in the weight updates. This was done by quantizing all tensors to 16-b FX, except for the internal accumulators which were stored in a 32-b floating-point format. An important distinction our work makes is the circumvention of the overhead of implementing stochastic quantization (Hubara et al., 2016). Similarly, DoReFa-Net (Zhou et al., 2016) stores internal weight representations in 32-b FL, but quantizes the remaining tensors more aggressively. Thus arises the need to re-scale and re-compute in floating-point format, which our work avoids. Finally, Köster et al. (2017) suggests a new number format – Flexpoint – and were able to train neural networks using slightly 16-b per tensor element, with 5 shared exponent bits and a per-tensor dynamic range tracking algorithm. Such tracking causes a hardware overhead bypassed by our work since the arithmetic is purely FX. Augmenting Flexpoint with stochastic quantization effectively results in WAGE (Wu et al., 2018b), and enables integer quantization of each tensor.

As seen above, none of the prior works address the problem of predicting precision requirements of all training signals. Furthermore, the choice of precision is made in an ad-hoc manner. In contrast, we propose a *systematic methodology* to determine *close-to-minimal* precision requirements for FX-only training of deep neural networks.

## 5.2 CONCLUSION

In this paper, we have presented a study of precision requirements in a typical back-propagation based training procedure of neural networks. Using a set of quantization criteria, we have presented a precision assignment methodology for which FX training is made statistically similar to the FL baseline, known to converge a priori. We realized FX training of four networks on the CIFAR-10, CIFAR-100, and SVHN datasets and quantified the associated complexity reduction gains in terms costs of training. We also showed that our precision assignment is nearly minimal.

The presented work relies on the statistics of all tensors being quantized *during training*. This necessitates an initial baseline run in floating-point which can be costly. An open problem is to predict a suitable precision configuration by only observing the data statistics and the network architecture. Future work can leverage the analysis presented in this paper to enhance the effectiveness of other network complexity reduction approaches. For instance, weight pruning can be viewed as a coarse quantization process (quantize to zero) and thus can potentially be done in a targeted manner by leveraging the information provided by noise gains. Furthermore, parameter sharing and clustering can be viewed as a form of vector quantization which presents yet another opportunity to leverage our method for complexity reduction.

## ACKNOWLEDGMENT

This work was supported in part by C-BRIC, one of six centers in JUMP, a Semiconductor Research Corporation (SRC) program sponsored by DARPA.

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

**Supplementary Material**

## A    SUMMARY OF QUANTIZATION SETUP

The quantization setup depicted in Figure 1 is summarized as follows:

- Feedforward computation at layer $l$:

$$A_{l+1} = f_l(A_l, W_l)$$

  where $f_l()$ is the function implemented at layer $l$, $A_l$ ($A_{l+1}$) is the activation tensor at layer $l$ ($l+1$) quantized to a normalized unsigned fixed-point format with precision $B_{A_l}$ ($B_{A_{l+1}}$), and $W_l$ is the weight tensor at layer $l$ quantized to a normalized signed fixed-point format with precision $B_{W_l}$. We further assume the use of a ReLU-like activation function with a clipping level of 2 and a max-norm constraint on the weights which are clipped between $[-1, 1]$ at every iteration.
- Back-propagation of activation gradients at layer $l$:

$$G_l^{(A)} = g_l^{(A)}(W_l, G_{l+1}^{(A)})$$

  where $g_l()^{(A)}$ is the function that back-propagates the activation gradients at layer $l$, $G_l^{(A)}$ ($G_{l+1}^{(A)}$) is the activation gradient tensor at layer $l$ ($l+1$) quantized to a signed fixed-point format with precision $B_{G_l^{(A)}}$ ($B_{G_{l+1}^{(A)}}$).
- Back-propagation of weight gradient tensor $G_l^{(W)}$ at layer $l$:

$$G_l^{(W)} = g_l^{(W)}(A_l, G_{l+1}^{(A)})$$

  where $g_l^{(W)}()$ is the function that back-propagates the weight gradients at layer $l$, and $G_l^{(W)}$ is quantized to a signed fixed-point format with precision $B_{G_l^{(W)}}$.
- Internal weight accumulator update at layer $l$:

$$W_l^{(acc)} = U(W_l^{(acc)}, G_l^{(W)}, \gamma)$$

  where $U()$ is the update function, $\gamma$ is the learning rate, and $W_l^{(acc)}$ is the internal weight accumulator tensor at layer $l$ quantized to signed fixed-point with precision $B_{W_l^{(acc)}}$. Note that, for the next iteration, $W_l$ is directly obtained from $W_l^{(acc)}$ via quantization to $B_{W_l}$ bits.

## B  FURTHER EXPLANATIONS AND MOTIVATIONS BEHIND QUANTIZATION CRITERIA

**Criterion 1 (EFQN)** is used to ensure that all feedforward quantization noise sources contribute equally to the $p_m$ budget. Indeed, if one of the $2L$ reflected quantization noise variances from the feedforward tensors onto $p_m$, say $V_{W_i \to p_m}$ for $i \in \{1, \dots, L\}$, largely dominates all others, it would imply that all tensors but $W_i$ are overly quantized. It would therefore be necessary to either increase the precision of $W_i$ or decrease the precisions of all other tensors. The application of Criterion 1 (EFQN) through the closed form expression (1) in Claim 1 solves this issue avoiding the need for a trial-and-error approach.

Because FX numbers require a constant PDR, clipping of gradients is needed since their dynamic range is arbitrary. Ideally, a very small PDR would be preferred in order to obtain quantization steps of small magnitude, and hence less quantization noise. We can draw parallels from signal processing theory, where it is known that for a given quantizer, the signal-to-quantization-noise ratio (SQNR) is equal to $SQNR(dB) = 6B + 4.78 - PAR$ where $PAR$ is the peak-to-average ratio, proportional to the PDR. Thus, we would like to reduce the PDR as much as possible in order to increase the SQNR for a given precision. However, this comes at the risk of overflows (due to clipping). **Criterion 2 (GC)** addresses this trade-off between quantization noise and overflow errors.

Since the back-propagation training procedure is an iterative one, it is important to ensure that any form of bias does not corrupt the weight update accumulation in a positive feedback manner. FX quantization, being a uniform one, is likely to induce such bias when quantized quantities, most notable gradients, are not uniformly distributed. **Criterion 3 (RQB)** addresses this issue by using $\eta$ as proxy to this bias accumulation a function of quantization step size and ensuring that its worst case value is small in magnitude.

**Criterion 4 (BQN)** is in fact an extension of Criterion 1 (EFQN), but for the back-propagation phase. Indeed, once the precision (and hence quantization noise) of weight gradients is set as per Criterion 3 (RQB), it is needed to ensure that the quantization noise source at the activation gradients would not contribute more noise to the updates. This criterion sets the quantization step of the activation gradients.

**Criterion 5 (AS)** ties together feedforward and gradient precisions through the weight accumulators. It is required to increment/decrement the feedforward weights whenever the accumulated updates cross-over the weight quantization threshold. This is used to set the PDR of the weight accumulators. Furthermore, since the precision of weight gradients has already been designed to account for quantization noise (through Criteria 2-4), the criterion requires that the accumulators do not cause additional noise.

## C   PROOF OF CLAIM 1

The validity of Claim 1 is derived from the following five lemmas. Note that each lemma addresses the satisfiability of one of the five quantization criteria presented in the main text and corresponds to part of Claim 1.

**Lemma 1.** *The EFQN criterion holds if the precisions $B_{W_l}$ and $B_{A_l}$ are set as follows:*

$$B_{W_l} = rnd\left(\log_2\left(\sqrt{\frac{E_{W_l \to p_m}}{E^{(\min)}}}\right)\right) + B^{(\min)} \;\;\&\;\; B_{A_l} = rnd\left(\log_2\left(\sqrt{\frac{E_{A_l \to p_m}}{E^{(\min)}}}\right)\right) + B^{(\min)}$$

*for $l = 1 \ldots L$, where $rnd()$ denotes the rounding operation, $B^{(\min)}$ is a reference minimum precision, and $E^{(\min)}$ is given by:*

$$E^{(\min)} = \min\left(\{E_{W_l \to p_m}\}_{l=1}^L, \{E_{A_l \to p_m}\}_{l=1}^L\right). \tag{5}$$

*Proof.* By definition of the reflected quantization noise variance, the EFQN, by definition, is satisfied if:

$$\frac{\Delta_{W_1}^2}{12} E_{W_1 \to p_m} = \ldots = \frac{\Delta_{W_L}^2}{12} E_{W_L \to p_m} = \frac{\Delta_{A_1}^2}{12} E_{A_1 \to p_m} = \ldots = \frac{\Delta_{A_L}^2}{12} E_{A_L \to p_m},$$

where the quantization noise gains are given by:

$$E_{W_l \to p_m} = \mathbb{E}\left[\sum_{\substack{i=1 \\ i \neq \hat{Y}_{fl}}}^M \frac{\sum_{w \in W_l}\left|\frac{\partial(Z_i - Z_{\hat{Y}_{fl}})}{\partial w}\right|^2}{2|Z_i - Z_{\hat{Y}_{fl}}|^2}\right] \;\;\&\;\; E_{A_l \to p_m} = \mathbb{E}\left[\sum_{\substack{i=1 \\ i \neq \hat{Y}_{fl}}}^M \frac{\sum_{a \in A_l}\left|\frac{\partial(Z_i - Z_{\hat{Y}_{fl}})}{\partial a}\right|^2}{2|Z_i - Z_{\hat{Y}_{fl}}|^2}\right] \tag{6}$$

for $l = 1 \ldots L$, where $\{Z_i\}_{i=1}^M$ are the soft outputs and $Z_{\hat{Y}_{fl}}$ is the soft output corresponding to $\hat{Y}_{fl}$. The expressions for these quantization gains are obtained by linearly expanding (across layers) those used in (Sakr et al., 2017). Note that a second order upper bound is used as a surrogate expression for $p_m$.

From the definition of quantization step size, the above is equivalent to:

$$2^{-2B_{W_1}} E_{W_1 \to p_m} = \ldots = 2^{-2B_{W_L}} E_{W_L \to p_m} = 2^{-2B_{A_1}} E_{A_1 \to p_m} = \ldots = 2^{-2B_{A_L}} E_{A_L \to p_m}.$$

Let $E^{(\min)}$ be as defined in (5):

$$E^{(\min)} = \min\left(\{E_{W_l \to p_m}\}_{l=1}^L, \{E_{A_l \to p_m}\}_{l=1}^L\right).$$

We can divide each term by $E^{(\min)}$:

$$2^{-2B_{W_1}} \frac{E_{W_1 \to p_m}}{E^{(\min)}} = \ldots = 2^{-2B_{W_L}} \frac{E_{W_L \to p_m}}{E^{(\min)}} = 2^{-2B_{A_1}} \frac{E_{A_1 \to p_m}}{E^{(\min)}} = \ldots = 2^{-2B_{A_L}} \frac{E_{A_L \to p_m}}{E^{(\min)}}$$

where each term is positive, so that we can take square roots and logarithms such that:

$$B_{W_1} - \log_2\left(\sqrt{\frac{E_{W_1 \to p_m}}{E^{(\min)}}}\right) = \ldots = B_{W_L} - \log_2\left(\sqrt{\frac{E_{W_L \to p_m}}{E^{(\min)}}}\right)$$

$$= B_{A_1} - \log_2\left(\sqrt{\frac{E_{A_1 \to p_m}}{E^{(\min)}}}\right) = \ldots = B_{A_L} - \log_2\left(\sqrt{\frac{E_{A_L \to p_m}}{E^{(\min)}}}\right)$$

Thus we equate all of the above to a reference precision $B^{(\min)}$ yielding:

$$B_{W_l} = \log_2\left(\sqrt{\frac{E_{W_l \to p_m}}{E^{(\min)}}}\right) + B^{(\min)} \;\;\&\;\; B_{A_l} = \log_2\left(\sqrt{\frac{E_{A_l \to p_m}}{E^{(\min)}}}\right) + B^{(\min)}$$

for $l = 1 \ldots L$. Note that because $E^{(\min)}$ is the least quantization noise gain, it is equal to one of the above quantization noise gains so that the corresponding precision actually equates $B^{(\min)}$. As precisions must be integer valued, each of $B^{(\min)}$, $\{B_{W_l}\}_{l=1}^L$, and $\{B_{A_l}\}_{l=1}^L$ have to be integers, and thus a rounding operation is to be applied on all logarithm terms. Doing so results in (1) from Lemma 1 which completes this proof. □

**Lemma 2.** *The GC criterion holds for $\beta_0 = 5\%$ provided the weight and activation gradients pre-defined dynamic ranges (PDRs) are lower bounded as follows:*

$$r_{G_l^{(W)}} \geq 2\sigma_{G_l^{(W)}}^{(\max)} \quad \& \quad r_{G_{l+1}^{(A)}} \geq 4\sigma_{G_{l+1}^{(A)}}^{(\max)} \quad for \ l = 1 \dots L$$

*where $\sigma_{G_l^{(W)}}^{(\max)}$ and $\sigma_{G_{l+1}^{(A)}}^{(\max)}$ are the largest ever recorded estimates of the weight and activation gradients standard deviations $\sigma_{G_l^{(W)}}$ and $\sigma_{G_{l+1}^{(A)}}$, respectively.*

*Proof.* Let us consider the case of weight gradients. The GC criterion, by definition requires:

$$\beta_{G_l^{(W)}} = \Pr\left(\left\{|g| \geq r_{G_l^{(W)}} : g \in G_l^{(W)}\right\}\right) < 0.05$$

Typically, weight gradients are obtained by computing the derivatives of a loss function with respect to a mini-batch. By linearity of derivatives, weight gradients are themselves averages of instantaneous derivatives and are hence expected to follow a Gaussian distribution by application of the Central Limit Theorem. Furthermore, the gradient mean was estimated during baseline training and was found to oscillate around zero.

Thus

$$\beta_{G_l^{(W)}} = 2Q\left(\frac{r_{G_l^{(W)}}}{\sigma_{G_l^{(W)}}}\right)$$

where we used the fact that a Gaussian distribution is symmetric and $Q()$ is the elementary Q-function, which is a decreasing function. Thus, in the worst case, we have:

$$\beta_{G_l^{(W)}} \leq 2Q\left(\frac{r_{G_l^{(W)}}}{\sigma_{G_l^{(W)}}^{(\max)}}\right).$$

Hence, for a PDR as suggested by the lower bound in (2):

$$r_{G_l^{(W)}} \geq 2\sigma_{G_l^{(W)}}^{(\max)}$$

in Lemma 2, we obtain the upper bound:

$$\beta_{G_l^{(W)}} \leq 2Q(2) = 0.044 < 0.05$$

which means the GC criterion holds and completes the proof.

For activation gradients, the same reasoning applies, but the choice of a larger PDR in (2):

$$r_{G_{l+1}^{(A)}} \geq 4\sigma_{G_{l+1}^{(A)}}^{(\max)}$$

than for weight gradients is due to the fact that the true dynamic range of the activation gradients is larger than the value indicated by the second moment. This stems from the use of activation functions such as ReLU which make the activation gradients sparse. We also recommend increasing the PDR even more when using regularizers that sparsify gradients such as Dropout (Srivastava et al., 2014) or Maxout (Goodfellow et al., 2013). □

**Lemma 3.** *The RQB criterion holds for $\eta_0 = 1\%$ provided the weight gradient quantization step size is upper bounded as follows:*

$$\Delta_{G_l^{(W)}} < \frac{\sigma_{G_l^{(W)}}^{(\min)}}{4} \quad for \ l = 1 \dots L$$

*where $\sigma_{G_l^{(W)}}^{(\min)}$ is the smallest ever recorded estimate of $\sigma_{G_l^{(W)}}$.*

*Proof.* For the Gaussian distributed (see proof of Lemma 2) weight gradient at layer $l$, the true mean conditioned on the first non-zero quantization region is given by:

$$\mu_{G_l^{(W)}} = \frac{\int_{\frac{\Delta_{G_l^{(W)}}}{2}}^{\frac{3\Delta_{G_l^{(W)}}}{2}} x \exp\left(-\frac{x^2}{2\sigma_{G_l^{(W)}}^2}\right) dx}{\left(Q\left(\frac{\Delta_{G_l^{(W)}}}{2\sigma_{G_l^{(W)}}}\right) - Q\left(\frac{3\Delta_{G_l^{(W)}}}{2\sigma_{G_l^{(W)}}}\right)\right)\sqrt{2\pi\sigma_{G_l^{(W)}}^2}}$$

$$= \frac{\sigma_{G_l^{(W)}}\left(\exp\left(-\frac{\Delta_{G_l^{(W)}}^2}{8\sigma_{G_l^{(W)}}^2}\right) - \exp\left(-\frac{9\Delta_{G_l^{(W)}}^2}{8\sigma_{G_l^{(W)}}^2}\right)\right)}{\left(Q\left(\frac{\Delta_{G_l^{(W)}}}{2\sigma_{G_l^{(W)}}}\right) - Q\left(\frac{3\Delta_{G_l^{(W)}}}{2\sigma_{G_l^{(W)}}}\right)\right)\sqrt{2\pi}},$$

where $\sigma_{G_l^{(W)}}$ is the standard deviation of $G_l^{(W)}$. By substituting $\Delta_{G_l^{(W)}} = \frac{\sigma_{G_l^{(W)}}}{4}$ into the above expression of $\mu_{G_l^{(W)}}$ and plugging in the definition of relative quantization bias, we obtain:

$$\eta_{G_l^{(W)}} = \frac{\left|\Delta_{G_l^{(W)}} - \mu_{G_l^{(W)}}\right|}{\mu_{G_l^{(W)}}} = 0.4\% < 1\%.$$

Hence, this choice of the quantization step satisfies the RQB. In order to ensure the RQB holds throughout training, $\sigma_{G_l^{(W)}}^{(\min)}$ is used in Lemma 3. This completes the proof. □

**Lemma 4.** *The BQN criterion holds provided the activation gradient quantization step size is upper bounded as follows:*

$$\Delta_{G_{l+1}^{(A)}} < \frac{\Delta_{G_l^{(W)}}}{\sqrt{\lambda_{G_{l+1}^{(A)} \to G_l^{(W)}}^{(\max)}}} \left(\frac{\left|G_l^{(W)}\right|}{\left|G_{l+1}^{(A)}\right|}\right)^{1/4} \qquad for\ l = 1\ldots L$$

*where $\lambda_{G_{l+1}^{(A)} \to G_l^{(W)}}^{(\max)}$, the largest singular value of the square-Jacobian (Jacobian matrix with squared entries) of $G_l^{(W)}$ with respect to $G_{l+1}^{(A)}$.*

*Proof.* Let us unroll $G_l^{(W)}$ and $G_{l+1}^{(A)}$ to vectors of size $\left|G_l^{(W)}\right|$ and $\left|G_{l+1}^{(A)}\right|$, respectively. The element-wise quantization noise variance of each weight gradient is $\frac{\Delta_{G_l^{(W)}}^2}{12}$. Therefore we have:

$$V_{G^{(W)} \to \Sigma_l} = \left|G_l^{(W)}\right| \frac{\Delta_{G_l^{(W)}}^2}{12}.$$

The reflected quantization noise variance from an activation gradient $g_a \in G_{l+1}^{(A)}$ onto a weight gradient $g_w \in G_l^{(W)}$ is

$$\left|\frac{\partial g_w}{\partial g_a}\right|^2 \frac{\Delta_{G_{l+1}^{(A)}}^2}{12},$$

where cross products of quantization noise are neglected (Sakr et al., 2017). Hence, the reflected quantization noise variance element-wise from $G_{l+1}^{(A)}$ onto $G_l^{(W)}$ is given by:

$$\frac{\Delta_{G_{l+1}^{(A)}}^2}{12} J_{G_{l+1}^{(A)} \to G_l^{(W)}} \mathbf{1}_{\left|G_{l+1}^{(A)}\right|},$$

where $J_{G_{l+1}^{(A)} \to G_l^{(W)}}$ is the square-Jacobian of $G_l^{(W)}$ with respect to $G_{l+1}^{(A)}$ and $\mathbf{1}$ denotes the all one vector with size denoted by its subscript. Hence, we have:

$$
\begin{aligned}
V_{G_{l+1}^{(A)} \to \Sigma_l} &= \frac{\Delta_{G_{l+1}^{(A)}}^2}{12} \left( J_{G_{l+1}^{(A)} \to G_l^{(W)}} \mathbf{1}_{\left|G_{l+1}^{(A)}\right|} \right)^T \mathbf{1}_{\left|G_l^{(W)}\right|} \\
&\leq \frac{\Delta_{G_l^{(A)}}^2}{12} \left\| J_{G_{l+1}^{(A)} \to G_l^{(W)}} \mathbf{1}_{\left|G_{l+1}^{(A)}\right|} \right\| \left\| \mathbf{1}_{\left|G_l^{(W)}\right|} \right\| \\
&\leq \sqrt{\left|G_l^{(W)}\right|} \frac{\Delta_{G_{l+1}^{(A)}}^2}{12} \left\| J_{G_{l+1}^{(A)} \to G_l^{(W)}} \right\| \left\| \mathbf{1}_{\left|G_{l+1}^{(A)}\right|} \right\| \\
&\leq \lambda_{G_{l+1}^{(A)} \to G_l^{(W)}}^{(\max)} \sqrt{\left|G_{l+1}^{(A)}\right| \left|G_l^{(W)}\right|} \frac{\Delta_{G_{l+1}^{(A)}}^2}{12},
\end{aligned}
$$

where we used the Cauchy-Schwarz inequality and the spectral norm of a matrix. Next we set this upper bound on $V_{G_{l+1}^{(A)} \to \Sigma_l}$ to be less than the value of $V_{G_l^{(W)} \to \Sigma_l}$ determined above. This condition, by definition, is enough to satisfy the BQN criterion. Rearranging terms yields Lemma 4 which completes the proof.

In the main text, it was menitoned that each entry in the Jacobian matrix above is already computed by the back-propagation algorithm. We now explain how. Let us denote the instantaneous loss function being minimized by $\xi$. Note that each entry of $J_{G_{l+1}^{(A)} \to G_l^{(W)}}$ is of the form $\left| \frac{\partial g_w}{\partial g_a^{(0)}} \right|^2$ where $g_w = \frac{\partial \xi}{\partial w}$ with $w \in W_l$ and $g_a^{(0)} = \frac{\partial \xi}{\partial a^{(0)}}$ with $a^{(0)} \in A_{l+1}$. The back-propagation algorithm computes $g_w$ using the chain rule as follows:

$$
g_w = \frac{\partial \xi}{\partial w} = \sum_{a^{(i)} \in A_{l+1}} \frac{\partial \xi}{\partial a^{(i)}} \frac{\partial a^{(i)}}{\partial w}.
$$

In particular, note that $g_a^{(0)}$ appears only once in the summation above and is multiplied by $\frac{\partial a^{(0)}}{\partial w}$. Thus $\frac{\partial g_w}{\partial g_a^{(0)}} = \frac{\partial a^{(0)}}{\partial w}$. This establishes that each entry of the Jacobian matrix is already computed via the back-propagation algorithm. $\qquad \square$

**Lemma 5.** *The AS criterion holds provided the accumulator PDR and quantization step size satisfy:*

$$
r_{W_l^{(acc)}} \geq 2^{-B_{W_l}} \quad \& \quad \Delta_{W_l^{(acc)}} < \gamma^{(\min)} \Delta_{G_l^{(W)}} \quad for \; l = 1 \ldots L
$$

*where $\gamma^{(\min)}$ is the smallest value of the learning rate used during training.*

*Proof.* The lower bound on the PDR of the weight accumulator, given by

$$
r_{W_l^{(acc)}} \geq 2^{-B_{W_l}}
$$

for $l = 1 \ldots L$, ensures that updates are able to cross over the feedforward weight quantization threshold so that it can be updated. Additionally, the lower bound on the quantization step size, given by

$$
\Delta_{W_l^{(acc)}} < \gamma^{(\min)} \Delta_{G_l^{(W)}}
$$

for $l = 1 \ldots L$, simply ensures that the internal weight accumulator overlaps with the least significant part of the representation of the weight gradient multiplied by the learning rate. Thus, the quantization noise of the internal accumulator is zero, or equivalently,

$$
V_{W_l^{(acc)} \to \Sigma_l^{(acc)}} = 0 \quad \text{for } l = 1 \ldots L
$$

which, by definition, is enough for the AS criterion to hold. Note that this criterion applies to the Vanilla-SGD learning rule (which was used in our experiments). Future work includes extending this criterion to other learning rules such as momentum and ADAM. $\qquad \square$

We close this appendix by discussing the approximation made by invoking the Central Limit Theorem (CLT) in the proofs of Lemmas 2 & 3. This approximation was made because, typically, a back-propagation iteration computes gradients of a loss function being averaged over a mini-batch of samples. By linearity of derivatives, the gradients themselves are averages, which warrants the invocation of the CLT. However, the CLT is an asymptotic result which might be imprecise for a finite number of samples. In typical training of neural networks, the number of samples, or mini-batch size, is in the range of hundreds or thousands (Goyal et al., 2017). It is therefore important to quantify the preciseness, or lack thereof, of the CLT approximation. On way to do so is via the Berry-Essen Theorem which considers the average of $n$ independent, identically distributed random variables with finite absolute third moment $\rho$ and standard deviation $\sigma$. The worst case deviation of the cumulative distribution of the true average from the of the approximated Gaussian random variable (via the CLT), also known as the Kolmogorov-Smirnov distance, $KS$, is upper bounded as follows: $KS < \frac{C\rho}{\sqrt{n}\sigma^3}$, where $C < 0.4785$ (Tyurin, 2010). Observe that the quantity $\frac{\rho}{\sigma^3}$ is data dependent. To estimate this quantity, we performed a forward-backward pass for all training samples at the start of each epoch for our four networks considered. The statistics $\rho$ and $\sigma$ were estimated by spatial (over tensors) and sample (over training samples) averages. The maximum value of the ratio $\frac{\rho}{\sigma^3}$ for all gradient tensors was found to be 2.8097. The mini-batch size we used in all our experiments was 256. Hence, we claim that the CLT approximation in Lemmas 2 & 3 is valid in our context up to a worst case Kolmogorov-Smirnov distance of $KS < \frac{0.4785 \times 2.8097}{\sqrt{256}} = 0.084$.

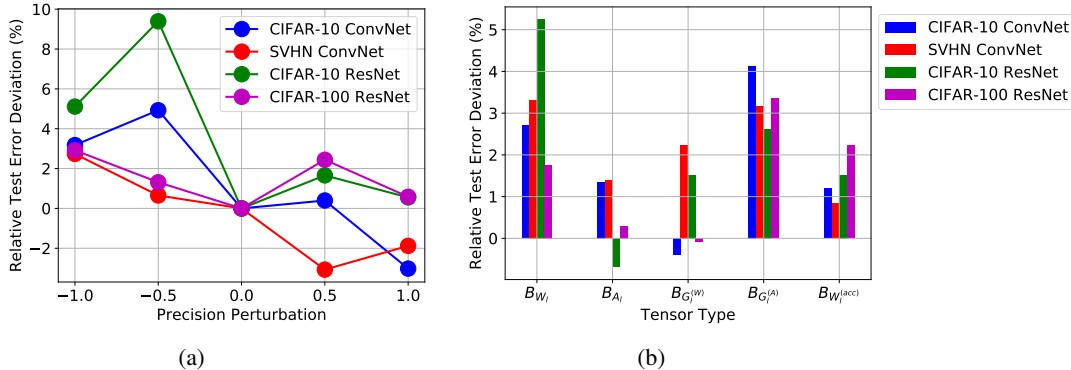

Figure 4: Additional experiments on minimality and sensitivity of $C_o$: relative test error deviation with respect to $C_o$ as a function of (a) random fractional precision perturbations, and (b) 1-b precision reduction per tensor type.

# D    ADDITIONAL RESULTS ON THE MINIMALITY AND SENSITIVITY OF $C_o$

The minimality experiments in the main paper only consider a full 1-b perturbation to the full precision configuration matrix. We further investigate the minimality of $C_o$ and its sensitivity to precision perturbation per tenor type. The results of this investigation are presented in Fig. 4.

First, we consider random fractional precision perturbations, meaning perturbations to the precision configuration matrix where only a random fraction $p$ of the $5L$ precision assignments is incremented or decremented. A fractional precision perturbation of 1 (-1) corresponds to $C_{+1}$ ($C_{-1}$). A fractional precision perturbation of 0.5 (-0.5) means that a randomly chosen half of the precision assignments is incremented (decremented). Figure 4 (a) shows the relative test error deviation compared to the test error associated with $C_o$ for various fractional precision perturbations. The error deviation is taken in a relative fashion to account for the variability of the different networks' accuracies. For instance, an absolute 1% difference in accuracy on a network trained on SVHN is significantly more severe than one on a network trained on CIFAR-100. It is observed that for negative precision perturbations the variation in test error is more important than for the case of positive perturbations. This is further encouraging evidence that $C_o$ is nearly minimal, in that a negative perturbation causes significant accuracy degradation while a positive one offers diminishing returns.

It is also interesting to study which of the $5L$ tensor types is most sensitive to precision reduction. To do so, we perform a similar experiment whereby we selectively decrement the precision of all tensors belonging to the same type (weights, activations, weight gradients, activation gradients, weight accumulators). The results of this experiment are found in Fig. 4 (b). It is found that the most sensitive tensor types are weights and activation gradients while the least sensitive ones are activations and weight gradients. This is an interesting finding raising further evidence that there exists some form of duality between the forward propagation of activations and back propagation of derivatives as far as numerical precision is concerned.

# E    ILLUSTRATION OF METHODOLOGY USAGE

We illustrate a step by step description of the application of our precision assignment methodology to the four networks we report results on.

## E.1    CIFAR-10 CONVNET

**Feedforward Precisions:** The first step in our methodology consists of setting the feedforward precisions $B_{W_l}$ and $B_{A_l}$. As per Claim 1, this requires using (1). To do so, it is first needed to compute the quantization noise gains using (6). Using the converged weights from the baseline run we obtain:

| Layer Index $l$ | 1 | 2 | 3 | 4 | 5 |
|---|---|---|---|---|---|
| $E_{W_l \to p_m}$ | 1.52E+06 | 1.24E+06 | 4.21E+06 | 3.57E+06 | 2.35E+06 |
| $E_{A_l \to p_m}$ | 5.51E+04 | 3.27E+02 | 5.15E+02 | 6.60E+02 | 7.78E+02 |
| Layer Index $l$ | 6 | 7 | 8 | 9 | |
| $E_{W_l \to p_m}$ | 5.61E+05 | 5.97E+04 | 3.23E+04 | 8.66E+03 | |
| $E_{A_l \to p_m}$ | 7.49E+02 | 6.32E+02 | 2.37E+02 | 9.47E+01 | |

And therefore, $E_{(\mathrm{min})} = 94.7$ and the feedforward precisions should be set according to (1) as follows:

| Layer Index $l$ | 1 | 2 | 3 | 4 | 5 |
|---|---|---|---|---|---|
| $B_{W_l}$ | $7+B^{(\mathrm{min})}$ | $7+B^{(\mathrm{min})}$ | $8+B^{(\mathrm{min})}$ | $8+B^{(\mathrm{min})}$ | $7+B^{(\mathrm{min})}$ |
| $B_{A_l}$ | $4+B^{(\mathrm{min})}$ | $1+B^{(\mathrm{min})}$ | $1+B^{(\mathrm{min})}$ | $1+B^{(\mathrm{min})}$ | $2+B^{(\mathrm{min})}$ |
| Layer Index $l$ | 6 | 7 | 8 | 9 | |
| $B_{W_l}$ | $6+B^{(\mathrm{min})}$ | $5+B^{(\mathrm{min})}$ | $4+B^{(\mathrm{min})}$ | $3+B^{(\mathrm{min})}$ | |
| $B_{A_l}$ | $1+B^{(\mathrm{min})}$ | $1+B^{(\mathrm{min})}$ | $1+B^{(\mathrm{min})}$ | $0+B^{(\mathrm{min})}$ | |

The value of $B^{(\mathrm{min})}$ is swept and $p_m$ i evaluated on the validation set. It is found that the smallest value of $B^{(\mathrm{min})}$ resulting in $p_m < 1\%$ is equal to 4 bits. Hence the feedforward precisions are set as follows and as illustrated in Figure 2:

| Layer Index $l$ | 1 | 2 | 3 | 4 | 5 | 6 | 7 | 8 | 9 |
|---|---|---|---|---|---|---|---|---|---|
| $B_{W_l}$ | 11 | 11 | 12 | 12 | 11 | 10 | 9 | 8 | 7 |
| $B_{A_l}$ | 8 | 5 | 5 | 5 | 6 | 5 | 5 | 5 | 4 |

**Gradient Precisions:** The second step of the methodology is to determine the precisions of weight $B_{G_l^{(W)}}$ and activation $B_{G_{l+1}^{(A)}}$ gradients. As per Claim 1, an important statistic is the spatial variance of the gradient tensors. We estimate these variances via moving window averages, where at each iteration, the running variance estimate $\hat{\sigma}^2$ is updated using the instantaneous variance $\tilde{\sigma}^2$ as follows:

$$\hat{\sigma}^2 \leftarrow (1-\theta)\hat{\sigma}^2 + \theta\tilde{\sigma}^2$$

where $\theta$ is the running average factor, chosen to be 0.1. The running variance estimate of each gradient tensor is dumped every epoch. Using the maximum recorded estimate and (2) we compute the PDRs of the gradient tensors (as a reminder, the PDR is forced to be a power of 2):

| Layer Index $l$ | 1 | 2 | 3 | 4 | 5 |
|---|---|---|---|---|---|
| $r_{G_l^{(W)}}$ | 5.00E-01 | 1.25E-01 | 1.25E-01 | 1.25E-01 | 6.25E-02 |
| $r_{G_{l+1}^{(A)}}$ | 4.88E-04 | 9.77E-04 | 9.77E-04 | 1.95E-03 | 7.81E-03 |
| Layer Index $l$ | 6 | 7 | 8 | 9 | |
| $r_{G_l^{(W)}}$ | 3.13E-02 | 3.13E-02 | 1.56E-02 | 1.25E-01 | |
| $r_{G_{l+1}^{(A)}}$ | 1.56E-02 | 7.81E-03 | 7.81E-03 | 3.13E-02 | |

Furthermore, using the minimum recorded estimates of the weight gradient spatial variances and (3) we compute the values of the quantization step sizes of the weight tensors:

| Layer Index $l$ | 1 | 2 | 3 | 4 | 5 |
|---|---|---|---|---|---|
| $\Delta_{G_l^{(W)}}$ | 3.91E-03 | 1.95E-03 | 9.77E-04 | 9.77E-04 | 4.88E-04 |

| Layer Index $l$ | 6 | 7 | 8 | 9 | |
|---|---|---|---|---|---|
| $\Delta_{G_l^{(W)}}$ | 2.44E-04 | 2.44E-04 | 1.22E-04 | 4.88E-04 | |

Hence the weight gradients precisions $B_{G_l^{(W)}}$ are set as follows and as illustrated in Figure 2:

| Layer Index $l$ | 1 | 2 | 3 | 4 | 5 | 6 | 7 | 8 | 9 |
|---|---|---|---|---|---|---|---|---|---|
| $B_{G_l^{(W)}}$ | 9 | 9 | 9 | 9 | 9 | 9 | 9 | 9 | 10 |

In order to compute the activation gradients precisions, (3) dictates that we need the values of largest singular values of the of the square-Jacobians of $G_l^{(W)}$ with respect to $G_{l+1}^{(A)}$ for $l = 1 \ldots L$. The square Jacobians matrices are estimated in a moving window fashion as for the variances above. However, instead of updating a matrix every iteration, the updates are done every first batch of every epoch. The following are the maximum recorded singluar values:

| Layer Index $l$ | 1 | 2 | 3 | 4 | 5 |
|---|---|---|---|---|---|
| $\lambda_{G_{l+1}^{(A)} \to G_l^{(W)}}^{(\max)}$ | 1.44E+02 | 2.37E+02 | 4.28E+02 | 2.03E+02 | 4.20E+01 |

| Layer Index $l$ | 6 | 7 | 8 | 9 | |
|---|---|---|---|---|---|
| $\lambda_{G_{l+1}^{(A)} \to G_l^{(W)}}^{(\max)}$ | 9.08E+00 | 1.37E+01 | 1.26E+01 | 3.51E+00 | |

Using the above values and (3) we obtain the values of the quantization step sizes for the activation gradients:

| Layer Index $l$ | 1 | 2 | 3 | 4 | 5 |
|---|---|---|---|---|---|
| $\Delta_{G_{l+1}^{(A)}}$ | 6.10E-05 | 3.05E-05 | 7.63E-06 | 1.53E-05 | 1.53E-05 |

| Layer Index $l$ | 6 | 7 | 8 | 9 | |
|---|---|---|---|---|---|
| $\Delta_{G_{l+1}^{(A)}}$ | 1.53E-05 | 1.53E-05 | 7.63E-06 | 6.10E-05 | |

Hence the activation gradients precisions $B_{G_{l+1}^{(A)}}$ are set as follows and as illustrated in Figure 2:

| Layer Index $l$ | 1 | 2 | 3 | 4 | 5 | 6 | 7 | 8 | 9 |
|---|---|---|---|---|---|---|---|---|---|
| $B_{G_{l+1}^{(A)}}$ | 5 | 8 | 9 | 9 | 11 | 12 | 11 | 11 | 11 |

**Internal Weight Accumulators Precisions:** By application of (4), we use the above results to obtain the internal weight accumulator precisions. The only additional information needed is the value of the smallest learning rate value used in the training, which in our case is 0.0001. We obtain the following precisions which are illustrated in Figure 2

| Layer Index $l$ | 1 | 2 | 3 | 4 | 5 | 6 | 7 | 8 | 9 |
|---|---|---|---|---|---|---|---|---|---|
| $B_{W_l^{(Acc)}}$ | 13 | 15 | 14 | 14 | 16 | 18 | 19 | 21 | 20 |

## E.2   SVHN CONVNET

**Feedforward Precisions:** The quantization noise gains are used to obtain values for the precisions as a function of $B^{(\min)}$ as summarized below:

| Layer Index $l$ | 1 | 2 | 3 | 4 | 5 |
|---|---|---|---|---|---|
| $E_{W_l \to p_m}$ | 3.07E+03 | 4.50E+02 | 1.54E+03 | 1.79E+03 | 6.01E+03 |
| $B_{W_l}$ | $6+B^{(\mathrm{min})}$ | $5+B^{(\mathrm{min})}$ | $6+B^{(\mathrm{min})}$ | $6+B^{(\mathrm{min})}$ | $7+B^{(\mathrm{min})}$ |
| $E_{A_l \to p_m}$ | 7.58E+02 | 2.86E+00 | 7.09E+00 | 2.55E+00 | 8.33E+00 |
| $B_{A_l}$ | $5+B^{(\mathrm{min})}$ | $1+B^{(\mathrm{min})}$ | $2+B^{(\mathrm{min})}$ | $1+B^{(\mathrm{min})}$ | $2+B^{(\mathrm{min})}$ |
| Layer Index $l$ | 6 | 7 | 8 | 9 | |
| $E_{W_l \to p_m}$ | 1.25E+03 | 7.91E+01 | 1.20E+01 | 9.13E+00 | |
| $B_{W_l}$ | $6+B^{(\mathrm{min})}$ | $4+B^{(\mathrm{min})}$ | $2+B^{(\mathrm{min})}$ | $2+B^{(\mathrm{min})}$ | |
| $E_{A_l \to p_m}$ | 8.18E+00 | 1.78E+01 | 1.14E+00 | 3.90E-01 | |
| $B_{A_l}$ | $2+B^{(\mathrm{min})}$ | $3+B^{(\mathrm{min})}$ | $1+B^{(\mathrm{min})}$ | $0+B^{(\mathrm{min})}$ | |

The value of $B^{(\mathrm{min})}$ is again swept, and it is found that the $p_m < 1\%$ for $B^{(\mathrm{min})} = 3$. The feedforward precisions are therefore set as follows and as illustrated in Figure 2:

| Layer Index $l$ | 1 | 2 | 3 | 4 | 5 | 6 | 7 | 8 | 9 |
|---|---|---|---|---|---|---|---|---|---|
| $B_{W_l}$ | 9 | 8 | 9 | 9 | 10 | 9 | 7 | 5 | 5 |
| $B_{A_l}$ | 8 | 4 | 5 | 4 | 6 | 6 | 7 | 4 | 3 |

**Gradient Precisions:** The spatial variance of the gradient tensors is used to determine the PDRs and the quantization step sizes of weight gradients. The singular values of the square-Jacobians are needed to determine the quantization step sizes of activation gradients. They were computed as follows:

| Layer Index $l$ | 1 | 2 | 3 | 4 | 5 |
|---|---|---|---|---|---|
| $r_{G_l^{(W)}}$ | 6.25E-02 | 1.56E-02 | 1.56E-02 | 1.56E-02 | 1.56E-02 |
| $\Delta_{G_l^{(W)}}$ | 2.44E-04 | 6.10E-05 | 6.10E-05 | 6.10E-05 | 6.10E-05 |
| $r_{G_{l+1}^{(A)}}$ | 4.88E-04 | 4.88E-04 | 9.77E-04 | 1.95E-03 | 3.91E-03 |
| $\lambda_{G_{l+1}^{(A)} \to G_l^{(W)}}^{(\mathrm{max})}$ | 5.13E+00 | 1.48E+02 | 3.25E+02 | 1.37E+02 | 8.84E+01 |
| $\Delta_{G_{l+1}^{(A)}}$ | 3.05E-05 | 9.54E-07 | 9.54E-07 | 9.54E-07 | 1.91E-06 |
| Layer Index $l$ | 6 | 7 | 8 | 9 | |
| $r_{G_l^{(W)}}$ | 7.81E-03 | 3.91E-03 | 3.91E-03 | 1.56E-02 | |
| $\Delta_{G_l^{(W)}}$ | 3.05E-05 | 1.53E-05 | 7.63E-06 | 6.10E-05 | |
| $r_{G_{l+1}^{(A)}}$ | 1.56E-02 | 3.91E-03 | 3.91E-03 | 3.13E-02 | |
| $\lambda_{G_{l+1}^{(A)} \to G_l^{(W)}}^{(\mathrm{max})}$ | 2.20E+01 | 9.58E+00 | 1.78E+00 | 1.71E+00 | |
| $\Delta_{G_{l+1}^{(A)}}$ | 1.91E-06 | 9.54E-07 | 9.54E-07 | 7.63E-06 | |

Hence the gradients precisions are set as follows and as illustrated in Figure 2:

| Layer Index $l$ | 1 | 2 | 3 | 4 | 5 | 6 | 7 | 8 | 9 |
|---|---|---|---|---|---|---|---|---|---|
| $B_{G_l^{(W)}}$ | 9 | 9 | 9 | 9 | 9 | 9 | 9 | 10 | 9 |
| $B_{G_{l+1}^{(A)}}$ | 5 | 10 | 11 | 12 | 12 | 14 | 13 | 13 | 13 |

**Internal Weight Accumulators Precisions:** The smallest learning rate value for this network is 0.001 which results in the following precisions for the internal weight accumulators as illustrated in Figure 2:

| Layer Index $l$ | 1 | 2 | 3 | 4 | 5 | 6 | 7 | 8 | 9 |
|---|---|---|---|---|---|---|---|---|---|
| $B_{W_l^{(Acc)}}$ | 14 | 17 | 16 | 16 | 15 | 17 | 20 | 23 | 20 |

### E.3 CIFAR-10 RESNET

**Feedforward Precisions:** The quantization noise gains are used to obtain values for the precisions as a function of $B^{(\mathrm{min})}$ as summarized below:

| Layer Index $l$ | 1 | 2 | 3 | 4 | 5 | 6 |
|---|---|---|---|---|---|---|
| $E_{W_l \to p_m}$ | 2.41E+03 | 9.80E+02 | 1.22E+03 | 1.62E+03 | 1.52E+03 | 3.05E+03 |
| $B_{W_l}$ | $11+B^{(\min)}$ | $10+B^{(\min)}$ | $10+B^{(\min)}$ | $10+B^{(\min)}$ | $10+B^{(\min)}$ | $11+B^{(\min)}$ |
| $E_{A_l \to p_m}$ | 7.32E-01 | 5.15E-01 | 1.29E-01 | 1.12E-01 | 7.31E-02 | 8.98E-02 |
| $B_{A_l}$ | $5+B^{(\min)}$ | $5+B^{(\min)}$ | $4+B^{(\min)}$ | $4+B^{(\min)}$ | $3+B^{(\min)}$ | $3+B^{(\min)}$ |
| Layer Index $l$ | 7 | 8 | 9 | 10 | 11 | 12 |
| $E_{W_l \to p_m}$ | 1.47E+03 | 2.15E+03 | 2.74E+03 | 4.96E+03 | 4.23E+03 | 4.20E+03 |
| $B_{W_l}$ | $10+B^{(\min)}$ | $11+B^{(\min)}$ | $11+B^{(\min)}$ | $11+B^{(\min)}$ | $11+B^{(\min)}$ | $12+B^{(\min)}$ |
| $E_{A_l \to p_m}$ | 7.70E-02 | 8.39E-02 | 6.38E-02 | 1.92E-01 | 1.54E-01 | 1.33E-01 |
| $B_{A_l}$ | $3+B^{(\min)}$ | $3+B^{(\min)}$ | $3+B^{(\min)}$ | $4+B^{(\min)}$ | $4+B^{(\min)}$ | $4+B^{(\min)}$ |
| Layer Index $l$ | 13 | 14 | 15 | 16 | 17 | 18 |
| $E_{W_l \to p_m}$ | 7.25E+03 | 2.99E+03 | 2.86E+03 | 3.00E+03 | 5.02E+03 | 4.34E+03 |
| $B_{W_l}$ | $11+B^{(\min)}$ | $11+B^{(\min)}$ | $11+B^{(\min)}$ | $11+B^{(\min)}$ | $10+B^{(\min)}$ | $10+B^{(\min)}$ |
| $E_{A_l \to p_m}$ | 1.13E-01 | 8.51E-02 | 6.57E-02 | 1.29E-01 | 6.51E-02 | 2.16E-02 |
| $B_{A_l}$ | $4+B^{(\min)}$ | $3+B^{(\min)}$ | $3+B^{(\min)}$ | $4+B^{(\min)}$ | $3+B^{(\min)}$ | $2+B^{(\min)}$ |
| Layer Index $l$ | 19 | 20 | 21 | 22 | | |
| $E_{W_l \to p_m}$ | 1.41E+03 | 1.30E+03 | 1.08E+02 | 8.31E+00 | | |
| $B_{W_l}$ | $9+B^{(\min)}$ | $7+B^{(\min)}$ | $11+B^{(\min)}$ | $11+B^{(\min)}$ | | |
| $E_{A_l \to p_m}$ | 4.80E-03 | 7.82E-04 | | | | |
| $B_{A_l}$ | $1+B^{(\min)}$ | $0+B^{(\min)}$ | | | | |

Note that for weights, layer depths 21 and 22 correspond to the strided convolutions in the shortcut connections of residual blocks 4 and 7, respectively. The value of $B^{(\min)}$ is again swept, and it is found that the $p_m < 1\%$ for $B^{(\min)} = 3$. The feedforward precisions are therefore set as follows and as illustrated in Figure 2:

| Layer Index $l$ | 1 | 2 | 3 | 4 | 5 | 6 | 7 | 8 | 9 | 10 | 11 |
|---|---|---|---|---|---|---|---|---|---|---|---|
| $B_{W_l}$ | 14 | 13 | 13 | 13 | 13 | 14 | 13 | 14 | 14 | 14 | 14 |
| $B_{A_l}$ | 8 | 8 | 7 | 7 | 6 | 6 | 6 | 6 | 6 | 7 | 7 |
| Layer Index $l$ | 12 | 13 | 14 | 15 | 16 | 17 | 18 | 19 | 20 | 21 | 22 |
| $B_{W_l}$ | 15 | 14 | 14 | 14 | 14 | 13 | 13 | 12 | 10 | 14 | 14 |
| $B_{A_l}$ | 7 | 7 | 6 | 6 | 7 | 6 | 5 | 4 | 3 | | |

**Gradient Precisions:** The spatial variance of the gradient tensors is used to determine the PDRs and the quantization step sizes of weight gradients. The singular values of the square-Jacobians are needed to determine the quantization step sizes of activation gradients. They were computed as follows:

| Layer Index $l$ | 1 | 2 | 3 | 4 | 5 | 6 |
|---|---|---|---|---|---|---|
| $r_{G_l^{(W)}}$ | 2.50E-01 | 6.25E-02 | 6.25E-02 | 3.13E-02 | 3.13E-02 | 3.13E-02 |
| $\Delta_{G_l^{(W)}}$ | 2.44E-04 | 3.05E-05 | 3.05E-05 | 3.05E-05 | 3.05E-05 | 3.05E-05 |
| $r_{G_{l+1}^{(A)}}$ | 4.88E-04 | 2.44E-04 | 2.44E-04 | 2.44E-04 | 2.44E-04 | 2.44E-04 |
| $\lambda^{(\max)}_{G_{l+1}^{(A)} \to G_l^{(W)}}$ | 8.07E+02 | 2.84E+03 | 2.84E+03 | 5.43E+03 | 5.43E+03 | 4.94E+03 |
| $\Delta_{G_{l+1}^{(A)}}$ | 7.63E-06 | 4.77E-07 | 4.77E-07 | 2.38E-07 | 2.38E-07 | 2.38E-07 |
| Layer Index $l$ | 7 | 8 | 9 | 10 | 11 | 12 |
| $r_{G_l^{(W)}}$ | 3.13E-02 | 3.13E-02 | 3.13E-02 | 3.13E-02 | 3.13E-02 | 3.13E-02 |
| $\Delta_{G_l^{(W)}}$ | 3.05E-05 | 3.05E-05 | 3.05E-05 | 3.05E-05 | 3.05E-05 | 3.05E-05 |
| $r_{G_{l+1}^{(A)}}$ | 2.44E-04 | 2.44E-04 | 4.88E-04 | 4.88E-04 | 2.44E-04 | 2.44E-04 |
| $\lambda^{(\max)}_{G_{l+1}^{(A)} \to G_l^{(W)}}$ | 4.94E+03 | 1.22E+03 | 1.22E+03 | 1.08E+03 | 1.08E+03 | 8.07E+02 |
| $\Delta_{G_{l+1}^{(A)}}$ | 2.38E-07 | 4.77E-07 | 4.77E-07 | 4.77E-07 | 4.77E-07 | 9.54E-07 |
| Layer Index $l$ | 13 | 14 | 15 | 16 | 17 | 18 |
| $r_{G_l^{(W)}}$ | 3.13E-02 | 3.13E-02 | 1.56E-02 | 1.56E-02 | 1.56E-02 | 1.56E-02 |
| $\Delta_{G_l^{(W)}}$ | 3.05E-05 | 3.05E-05 | 1.53E-05 | 1.53E-05 | 1.53E-05 | 1.53E-05 |
| $r_{G_{l+1}^{(A)}}$ | 2.44E-04 | 4.88E-04 | 4.88E-04 | 4.88E-04 | 2.44E-04 | 2.44E-04 |
| $\lambda^{(\max)}_{G_{l+1}^{(A)} \to G_l^{(W)}}$ | 8.07E+02 | 1.93E+02 | 1.93E+02 | 2.98E+02 | 2.98E+02 | 3.01E+02 |
| $\Delta_{G_{l+1}^{(A)}}$ | 9.54E-07 | 9.54E-07 | 9.54E-07 | 4.77E-07 | 2.38E-07 | 2.38E-07 |
| Layer Index $l$ | 19 | 20 | 21 | 22 | | |
| $r_{G_l^{(W)}}$ | 1.56E-02 | 1.56E-02 | 1.56E-02 | 2.50E-01 | | |
| $\Delta_{G_l^{(W)}}$ | 7.63E-06 | 7.63E-06 | 1.91E-06 | 3.05E-05 | | |
| $r_{G_{l+1}^{(A)}}$ | 2.44E-04 | 6.25E-02 | | | | |
| $\lambda^{(\max)}_{G_{l+1}^{(A)} \to G_l^{(W)}}$ | 3.01E+02 | 2.32E+01 | | | | |
| $\Delta_{G_{l+1}^{(A)}}$ | 5.96E-08 | 3.81E-06 | | | | |

Hence the gradients precisions are set as follows and as illustrated in Figure 2:

| Layer Index $l$ | 1 | 2 | 3 | 4 | 5 | 6 | 7 | 8 | 9 | 10 | 11 |
|---|---|---|---|---|---|---|---|---|---|---|---|
| $B_{G_l^{(W)}}$ | 11 | 12 | 12 | 11 | 11 | 11 | 11 | 11 | 11 | 11 | 11 |
| $B_{G_{l+1}^{(A)}}$ | 7 | 10 | 10 | 11 | 11 | 11 | 11 | 10 | 11 | 11 | 10 |
| Layer Index $l$ | 12 | 13 | 14 | 15 | 16 | 17 | 18 | 19 | 20 | 21 | 22 |
| $B_{G_l^{(W)}}$ | 11 | 11 | 11 | 11 | 11 | 11 | 11 | 12 | 12 | 14 | 14 |
| $B_{G_{l+1}^{(A)}}$ | 9 | 9 | 10 | 10 | 11 | 11 | 11 | 13 | 15 | | |

**Internal Weight Accumulators Precisions:** The smallest learning rate value for this network is 0.001 which results in the following precisions for the internal weight accumulators as illustrated in Figure 2:

| Layer Index $l$ | 1 | 2 | 3 | 4 | 5 | 6 | 7 | 8 | 9 | 10 | 11 |
|---|---|---|---|---|---|---|---|---|---|---|---|
| $B_{W_l^{(Acc)}}$ | 9 | 13 | 13 | 13 | 13 | 12 | 13 | 12 | 12 | 12 | 12 |
| Layer Index $l$ | 12 | 13 | 14 | 15 | 16 | 17 | 18 | 19 | 20 | 21 | 22 |
| $B_{W_l^{(Acc)}}$ | 11 | 12 | 13 | 13 | 13 | 15 | 15 | 18 | 16 | 12 | 13 |

### E.4  CIFAR-100 RESNET

**Feedforward Precisions:** The quantization noise gains are used to obtain values for the precisions as a function of $B^{(\min)}$ as summarized below:

| Layer Index $l$ | 1 | 2 | 3 | 4 | 5 | 6 |
|---|---|---|---|---|---|---|
| $E_{W_l \to p_m}$ | 2.32E+03 | 8.23E+02 | 1.18E+03 | 1.28E+03 | 1.70E+03 | 2.78E+03 |
| $B_{W_l}$ | $10+B^{(\min)}$ | $9+B^{(\min)}$ | $10+B^{(\min)}$ | $10+B^{(\min)}$ | $10+B^{(\min)}$ | $10+B^{(\min)}$ |
| $E_{A_l \to p_m}$ | 1.42E+00 | 7.84E-01 | 2.52E-01 | 1.46E-01 | 7.68E-02 | 7.40E-02 |
| $B_{A_l}$ | $5+B^{(\min)}$ | $4+B^{(\min)}$ | $4+B^{(\min)}$ | $3+B^{(\min)}$ | $3+B^{(\min)}$ | $3+B^{(\min)}$ |
| Layer Index $l$ | 7 | 8 | 9 | 10 | 11 | 12 |
| $E_{W_l \to p_m}$ | 3.03E+03 | 5.80E+03 | 7.29E+03 | 9.20E+03 | 9.81E+03 | 1.41E+04 |
| $B_{W_l}$ | $10+B^{(\min)}$ | $11+B^{(\min)}$ | $11+B^{(\min)}$ | $11+B^{(\min)}$ | $11+B^{(\min)}$ | $11+B^{(\min)}$ |
| $E_{A_l \to p_m}$ | 7.52E-02 | 8.70E-02 | 1.38E-01 | 2.49E-01 | 2.11E-01 | 1.51E-01 |
| $B_{A_l}$ | $3+B^{(\min)}$ | $3+B^{(\min)}$ | $3+B^{(\min)}$ | $4+B^{(\min)}$ | $3+B^{(\min)}$ | $3+B^{(\min)}$ |
| Layer Index $l$ | 13 | 14 | 15 | 16 | 17 | 18 |
| $E_{W_l \to p_m}$ | 7.67E+03 | 1.40E+04 | 1.13E+04 | 1.09E+04 | 5.35E+03 | 3.97E+03 |
| $B_{W_l}$ | $11+B^{(\min)}$ | $11+B^{(\min)}$ | $11+B^{(\min)}$ | $11+B^{(\min)}$ | $11+B^{(\min)}$ | $11+B^{(\min)}$ |
| $E_{A_l \to p_m}$ | 1.54E-01 | 1.09E-01 | 1.93E-01 | 2.36E-01 | 1.27E-01 | 3.01E-02 |
| $B_{A_l}$ | $3+B^{(\min)}$ | $3+B^{(\min)}$ | $3+B^{(\min)}$ | $4+B^{(\min)}$ | $3+B^{(\min)}$ | $2+B^{(\min)}$ |
| Layer Index $l$ | 19 | 20 | 21 | 22 | | |
| $E_{W_l \to p_m}$ | 8.35E+02 | 2.30E+01 | 6.78E+03 | 6.03E+03 | | |
| $B_{W_l}$ | $9+B^{(\min)}$ | $7+B^{(\min)}$ | $11+B^{(\min)}$ | $11+B^{(\min)}$ | | |
| $E_{A_l \to p_m}$ | 2.01E-02 | 1.80E-03 | | | | |
| $B_{A_l}$ | $2+B^{(\min)}$ | $0+B^{(\min)}$ | | | | |

The value of $B^{(\min)}$ is again swept, and it is found that the $p_m < 1\%$ for $B^{(\min)} = 3$. The feedforward precisions are therefore set as follows and as illustrated in Figure 2:

| Layer Index $l$ | 1 | 2 | 3 | 4 | 5 | 6 | 7 | 8 | 9 | 10 | 11 |
|---|---|---|---|---|---|---|---|---|---|---|---|
| $B_{W_l}$ | 13 | 12 | 13 | 13 | 13 | 13 | 13 | 14 | 14 | 14 | 14 |
| $B_{A_l}$ | 8 | 7 | 7 | 6 | 6 | 6 | 6 | 6 | 6 | 7 | 6 |
| Layer Index $l$ | 12 | 13 | 14 | 15 | 16 | 17 | 18 | 19 | 20 | 21 | 22 |
| $B_{W_l}$ | 14 | 14 | 14 | 14 | 14 | 14 | 14 | 12 | 10 | 14 | 14 |
| $B_{A_l}$ | 6 | 6 | 6 | 6 | 7 | 6 | 5 | 5 | 3 | | |

**Gradient Precisions:** The spatial variance of the gradient tensors is used to determine the PDRs and the quantization step sizes of weight gradients. The singular values of the square-Jacobians are needed to determine the quantization step sizes of activation gradients. They were computed as follows:

| Layer Index $l$ | 1 | 2 | 3 | 4 | 5 | 6 |
|---|---|---|---|---|---|---|
| $r_{G_l^{(W)}}$ | 5.00E-01 | 6.25E-02 | 6.25E-02 | 3.13E-02 | 6.25E-02 | 3.13E-02 |
| $\Delta_{G_l^{(W)}}$ | 6.10E-05 | 1.53E-05 | 1.53E-05 | 1.53E-05 | 1.53E-05 | 1.53E-05 |
| $r_{G_{l+1}^{(A)}}$ | 2.44E-04 | 1.22E-04 | 6.10E-05 | 6.10E-05 | 6.10E-05 | 6.10E-05 |
| $\lambda_{G_{l+1}^{(A)} \to G_l^{(W)}}^{(\max)}$ | 6.46E+02 | 1.86E+03 | 1.86E+03 | 3.54E+03 | 3.54E+03 | 5.11E+03 |
| $\Delta_{G_{l+1}^{(A)}}$ | 9.54E-07 | 1.19E-07 | 1.19E-07 | 1.19E-07 | 1.19E-07 | 5.96E-08 |
| Layer Index $l$ | 7 | 8 | 9 | 10 | 11 | 12 |
| $r_{G_l^{(W)}}$ | 3.13E-02 | 3.13E-02 | 3.13E-02 | 3.13E-02 | 3.13E-02 | 3.13E-02 |
| $\Delta_{G_l^{(W)}}$ | 1.53E-05 | 1.53E-05 | 1.53E-05 | 1.53E-05 | 1.53E-05 | 1.53E-05 |
| $r_{G_{l+1}^{(A)}}$ | 6.10E-05 | 1.22E-04 | 1.22E-04 | 1.22E-04 | 1.22E-04 | 1.22E-04 |
| $\lambda_{G_{l+1}^{(A)} \to G_l^{(W)}}^{(\max)}$ | 5.11E+03 | 1.05E+03 | 1.05E+03 | 8.23E+02 | 8.23E+02 | 6.37E+02 |
| $\Delta_{G_{l+1}^{(A)}}$ | 5.96E-08 | 1.19E-07 | 1.19E-07 | 2.38E-07 | 2.38E-07 | 2.38E-07 |
| Layer Index $l$ | 13 | 14 | 15 | 16 | 17 | 18 |
| $r_{G_l^{(W)}}$ | 3.13E-02 | 3.13E-02 | 1.56E-02 | 3.13E-02 | 1.56E-02 | 1.56E-02 |
| $\Delta_{G_l^{(W)}}$ | 1.53E-05 | 7.63E-06 | 7.63E-06 | 7.63E-06 | 7.63E-06 | 7.63E-06 |
| $r_{G_{l+1}^{(A)}}$ | 1.22E-04 | 1.22E-04 | 2.44E-04 | 1.22E-04 | 6.10E-05 | 6.10E-05 |
| $\lambda_{G_{l+1}^{(A)} \to G_l^{(W)}}^{(\max)}$ | 6.37E+02 | 2.31E+02 | 2.31E+02 | 2.79E+02 | 2.79E+02 | 2.80E+02 |
| $\Delta_{G_{l+1}^{(A)}}$ | 2.38E-07 | 2.38E-07 | 2.38E-07 | 1.19E-07 | 1.19E-07 | 1.19E-07 |
| Layer Index $l$ | 19 | 20 | 21 | 22 | | |
| $r_{G_l^{(W)}}$ | 1.56E-02 | 6.25E-02 | 3.13E-02 | 1.56E-02 | | |
| $\Delta_{G_l^{(W)}}$ | 3.81E-06 | 7.63E-06 | 1.53E-05 | 7.63E-06 | | |
| $r_{G_{l+1}^{(A)}}$ | 6.10E-05 | 7.81E-03 | | | | |
| $\lambda_{G_{l+1}^{(A)} \to G_l^{(W)}}^{(\max)}$ | 2.80E+02 | 7.81E+01 | | | | |
| $\Delta_{G_{l+1}^{(A)}}$ | 5.96E-08 | 2.38E-07 | | | | |

Hence the gradients precisions are set as follows and as illustrated in Figure 2:

| Layer Index $l$ | 1 | 2 | 3 | 4 | 5 | 6 | 7 | 8 | 9 | 10 | 11 |
|---|---|---|---|---|---|---|---|---|---|---|---|
| $B_{G_l^{(W)}}$ | 14 | 13 | 13 | 12 | 13 | 12 | 12 | 12 | 12 | 12 | 12 |
| $B_{G_{l+1}^{(A)}}$ | 9 | 11 | 10 | 10 | 10 | 11 | 11 | 11 | 11 | 10 | 10 |
| Layer Index $l$ | 12 | 13 | 14 | 15 | 16 | 17 | 18 | 19 | 20 | 21 | 22 |
| $B_{G_l^{(W)}}$ | 12 | 12 | 13 | 12 | 13 | 12 | 12 | 13 | 14 | 12 | 12 |
| $B_{G_{l+1}^{(A)}}$ | 10 | 10 | 10 | 11 | 11 | 10 | 10 | 11 | 16 | | |

**Internal Weight Accumulators Precisions:** The smallest learning rate value for this network is 0.001 which results in the following precisions for the internal weight accumulators as illustrated in Figure 2:

| Layer Index $l$ | 1 | 2 | 3 | 4 | 5 | 6 | 7 | 8 | 9 | 10 | 11 |
|---|---|---|---|---|---|---|---|---|---|---|---|
| $B_{W_l^{(Acc)}}$ | 12 | 15 | 14 | 14 | 14 | 14 | 14 | 13 | 13 | 13 | 13 |
| Layer Index $l$ | 12 | 13 | 14 | 15 | 16 | 17 | 18 | 19 | 20 | 21 | 22 |
| $B_{W_l^{(Acc)}}$ | 13 | 13 | 14 | 14 | 14 | 14 | 14 | 17 | 18 | 13 | 14 |

