# OpenReview forum: "Per-Tensor Fixed-Point Quantization of the Back-Propagation Algorithm"
_ICLR.cc/2019/Conference_

### Official Review · AnonReviewer3 · 2018-10-30
**Nice work**

**Rating:** 8
**Confidence:** 4

**Review:**

Summarization:
This paper presents a framework (called FX Network) of quantizing the weights and gradients of neural networks, based on five quantization criteria proposed in literature. The proposed framework can quantize the neural network obtaining a minimal or close-to-minimal error for a pre-specified precision level.


Pros:
- The proposed FX network can quantize all variables including both network weights and back-propagated gradients.
- Promising results have been obtained. Experimental results on CIFAR have shown that the proposed quantization framework had reduced the representational cost, computational cost, and the communication  by up to 6x, 8x, and 4x, respectively, compared to the 32-b FL baseline and related works.
- The paper is well written.




Cons:
- The experiment results showed in Figure 3 are quite confusing: why do the curves of the test error and loss suddenly drop at epoch 100? Explanation is needed.

- This proposed quantization method require to pre-train a network with high precision in advance, similarly as the student-teacher framework or knowledge distillation. Different from BN and TG, FX network requires to pre-train a 32-b floating-point network, which requires more extra computational costs.

- How does the quantization method compare with strategies like parameter pruning and sharing? It is better to see a discussion with them. It is also suggested to show the improvement of the proposed framework in terms of inference time during test.

---

> ### Author Response · Authors · 2018-11-08
> **Reply to AnonReviewer3**
>
> Dear AnonReviewer3,
>
> Thank you very much for your encouraging review and thoughtful comments! We will make revisions to address the several points you have raised in your review; however, we first wish to reply to your comments:
>
>
> -"why do the curves of the test error and loss suddenly drop at epoch 100?"
>
> -- This is actually not related to the thesis of our work. It is just part of the training procedure we have used, which includes a scheduled learning rate. In particular, at epoch 100, the learning rate is decreased by a factor of 10, which marks the switch between exploration and exploitation phases. Such behavior is very typical of neural network training.
>
>
> -"This proposed quantization method requires to pre-train a network with high precision in advance... which requires more extra computational costs"
>
> -- You are right. This assumption is mentioned Section 1, 4th paragraph. Such assumption has been made in the past, for instance by Sakr et al. (ICML’17, which we cited) for determining inference precisions. Since much of training is done in floating-point anyway (most notably, in order to establish a benchmarking baseline), our approach simply leverages data statistics to find a min precision network. We will make sure to raise this issue more explicitly in our revised version.
>
>
> -"How does the quantization method compare with strategies like parameter pruning and sharing"
>
> -- We have focused on quantization as it provides by itself a large body of work inside the general area of resource constrained machine learning. That is to say, in our paper, we have treated other methods such as parameter pruning and sharing as orthogonal. Interestingly, we do believe that other strategies are related and that we may use a similar approach as the presented method to address parameter pruning and sharing. In particular, we believe that one may perform a noise analysis, similar to our analysis for determining feedforward precisions, applied to the weights, where the noise value is equal to the weight magnitude. Such analysis potentially acts as a weight pruning criterion. Furthermore, parameter sharing, if obtained via some form of clustering algorithm, presents a setup of vector (nonlinear) quantization. Thus, we again believe that we may leverage our method to address this other strategy of complexity reduction. These are definitely interesting issues to consider for future work, and, as per your suggestion, we will include a discussion.

---

> > ### Author Response · Authors · 2018-11-22
> > **Revision Uploaded**
> >
> > Dear AnonReviewer3,
> >
> > This is a note to let you know that we have uploaded our revision. To make it easy for you to track changes, we have typed all modifications with respect to the original manuscript in the blue color.
> >
> > In particular, please note that we have addressed your two comments: In the introduction, we have raised the issue that our assumptions require us to pre-train in FL which can be costly. We have provided justification that such cost could be amortized, in line with our original response to your review. We have also included a discussion about other complexity reduction techniques in the conclusion as per your request.
> >
> > We thank you again for your nice review!

---

### Official Review · AnonReviewer2 · 2018-11-03
**Paper needs to be re-written**

**Rating:** 3
**Confidence:** 2

**Review:**

This papers introduces a quantization scheme for the back-propagation algorithm to reduce the bit size in the target neural networks. While the paper introduces one way to bring the quantization inside the training procedure and shows the tradeoff between number of bits and the accuracy, the paper is poorly written so it is hard to understand the paper's main proposal.
So I would recommend to re-organize the paper and introduce one toy example to illustrate how the proposed method works in the training time and the inference time.
Currently the important part, the overall architecture, is explained in the appendix, not in the main paper.
The main idea is rather simple, to introduce a quantizer in various components in the back-propagation algorithm.
I think we need a clear explanation on "how to" quantize each tensor in each quantizer, instead of many obscure terms in the section 2 and 3. Also the important numbers are in the appendix C, but their meanings are hard to understand.

Also in general, quantization is one way of reducing training and inference computational complexity. There are other ways of achieving the same purpose such as distillation to a smaller network (less parameters), etc, so in order to argue the computational gains over this obvious approach, we need a training time and inference time benchmark.

---

> ### Author Response · Authors · 2018-11-08
> **Reply to AnonReviewer2**
>
> Dear AnonReviewer2,
>
> Thank you for your review. We are sorry that you had difficulties reading our paper. We will make revisions to hopefully address your concerns; however, we first wish to reply to your comments:
>
>
>  - "I would recommend to re-organize the paper and introduce one toy example to illustrate how the proposed method works in the training time and the inference time"
>
> -- We have done that to a great extent. The method is obtained by application of Claim 1 (eq. (1) (2) (3) (4) - for every precision to be determined, there is a closed form expression to be used). At the end of Section 3, we provide practical considerations to keep in mind when using the method. When introducing the results at the start of Section 4, we mention that a step by step procedure is given in Appendix C to illustrate explicitly how the method is to be used. We will revise these parts to make the illustrations clearer.
>
>
>  - "Currently the important part, the overall architecture, is explained in the appendix, not in the main paper"
>
> -- The architecture is described in Figure 1. This figure conveys all of the information needed to understand our setup. We have included Appendix A as a supplement for readers not familiar with the back-propagation algorithm. The Appendix itself is by no means the important part, nor is it necessary, the paper, as is, very well stands on its own without it.
>
>
> - "I think we need a clear explanation on "how to" quantize each tensor in each quantizer, instead of many obscure terms in the section 2 and 3"
>
> -- The issue has been explained with great transparency in the paper. In Section 2.1, we explicitly define what a fixed-point number representation is, and what scalar values are representable. Determining the precision $B$ is enough to fully characterize the quantizer in the case of normalized scalars. Otherwise, the quantizer can be characterized once the values of the pre-defined dynamic range $r$ and quantization step $\Delta$ are set. Then, in Section 3, we derive closed form formulas for each of these quantities, for all tensors, and that is how we characterize all quantizers. This is actually even mentioned in the "practical considerations" paragraph following Claim 1.
>
>
> - "Also, the important numbers are in the appendix C, but their meanings are hard to understand"
>
> -- Appendix C contains intermediate results utilized to obtain the different precisions in the network. The latter are the important numbers, along with the convergence behaviors in Figure 3 and the several comparisons in Table 1. The role of Appendix C is purely illustratory. The appendices are supplementary material with details, proofs, and examples, meant to illustrate and reproduce our results. They are not required to understand the paper. The main paper stands on its own.
>
>
> -"There are other ways of achieving the same purpose such as distillation to a smaller network (less parameters), etc, so in order to argue the computational gains over this obvious approach, we need a training time and inference time benchmark"
>
> -- You are absolutely right. In fact, the general area of resource constrained machine learning has been growing rapidly in the past few years. Particularly, many works have addressed the problem of training/inference with reduced precision (see Section 5.1), and those naturally come closest to our work. It is for that reason that we have focused on this body of work in most of our analyses and comparisons. Of course, distillation techniques are also related and we will therefore include a comparative discussion.
>
>
> Finally, with all of the above in mind, we hope you would give us a second chance when updating your review.

---

> > ### Author Response · Authors · 2018-11-22
> > **Revision Uploaded**
> >
> > Dear AnonReviewer2,
> >
> > This is a note to let you know that we have uploaded our revision. To make it easy for you to track changes, we have typed all modifications with respect to the original manuscript in the blue color.
> >
> > We wish to let you know that we have tried to make all clarifications needed. Further, we included a supplementary discussion on the motivation behind our method (Page 11). We also included a discussion on other methods of complexity reduction in our conclusion.
> >
> > We would truly appreciate it if you could give our paper a second chance and update your review. Thank you very much.

---

> > > ### Comment · AnonReviewer2 · 2018-12-01
> > > **Many thanks to the revision, but still stick to my recommendation.**
> > >
> > > I read through the paper again, but my concern is not resolved. Note that the appendix is usually an optional to read. The main part of the paper should motivate the core idea and explain the concept, and the readers should not resort to the appendix to understand the main concept and the motivation behind it.
> > > I am not saying the paper's technical contribution is marginal. It contains an interesting idea, and can be a way to enable full FX training without FL pre-training.
> > > I just wanted to point out that
> > >
> > > 1. The "main" part of the paper should be clearly written so that the readers can easily comprehend the core idea and the motivation behind it.
> > > 2. Also, I still think it's important to compare the timing benchmark with other approaches such as a distillation, etc, or at least even trying to address why a distillation to a smaller network should not be compared to this approach in the main paper.
> > >
> > > so that the paper can be greatly improved in terms of the readability.

---

> > > > ### Author Response · Authors · 2018-12-02
> > > > **Response to AnonReviewer2**
> > > >
> > > > Dear AnonReviewer2,
> > > >
> > > > Thank you for going over our paper again and for acknowledging that our contribution is substantive.
> > > >
> > > > First, we wish to say that we absolutely agree with you that the main part of the paper should be self-contained and we emphasize that it is given the page limits. For example, Section 2 sets up the notation and defines the problem in a concise yet complete manner so that we can get to the main body of work as soon as possible. For the interested reader, the appendix elaborates on the set-up and provides a nuanced description. The appendix is not needed to understand the paper. Please understand that we have taken the matter of clear exposition seriously and worked hard to make the paper well-written before submitting to ICLR. Also, other reviewers have found our paper is clear and well-written.
> > > >
> > > > Second, as per our original reply to you and AnonReviewer3, and as revised in our manuscript, our comparisons have focused on quantization since a large body of work on this topic already exists. That is to say, in our paper, we have treated other methods such as distillation techniques as orthogonal in that they can be used additionally.
> > > >
> > > > This is an important piece of work for the ICLR community since it enables fixed-point training which is critical for both edge and cloud platforms. We respectfully urge you to reconsider your evaluation.

---

### Official Review · AnonReviewer4 · 2018-11-10
**the introduction of criteria 2-5 are kind of heuristic and lack of clarity**

**Rating:** 7
**Confidence:** 3

**Review:**

This paper proposes a FX (fixed point) framework to calculate the reduced bit numbers, which can (I) use float numbers with the reduced bits to represent each NN layer's weight values W_l, activation values A_l, gradients of weights G_l^W, gradients of nodes G_{l+1}^A, and the cumulated (updated) weight W_l^(acc); (II) with the reduced representations, the training loss and testing loss will not be sacrificed much comparing with the original FL framework, where each float number is 32-bit.

Some positive points:
(a) The proposed FX framework can reduce cost for both inference and training.
(b) The experimental results looks promising.
(c) The Criteria 1-5 seems systematic and the conditions in Claim 1 can be used to calculate the required bit numbers. The author proposed an implementation of Claim 1.

Some negative points / questions:

(a) The most important part of the paper is Criteria 1-5. Criteria 1 generalizes the idea from Sakr et al. 2017, to force the contributions of weight and activation almost at the same order, which seems reasonable to make the mismatch budget p_m smallest. But the other criteria (with their corresponding notions, e.g., Criterion 2 and the concept of clipping rate \beta) are introduced in a way which is not clear enough and make the audience confused. For example, why is clipping rate \beta and relative quantization bias \eta is needed here and what is their relationship with the usual weight gradient clip norm (5% target \beta and 1% \eta target correspond to what order of clip norm)? Criteria 4 & 5 are introduced in the same way with one sentence explained like heuristics. They seem to me are introduced just for reducing corresponding bit numbers. More motivation and explanation of introducing these criteria and notions are needed.

(b) For W_l and A_l, the necessary bit numbers are calculated using Criteria 1 & EFQN condition. But for gradients, their PDRs and Deltas are calculated using other criteria & conditions. Then how to calculate their bit numbers from PDRs and Deltas?

(c) The proof of Lemma 2 & 3 directly used CLT for mini-batch average gradient items. CLT is for asymptotic case and in finite sample case it is not true. So it is heuristic calculation rather than lemma with proof (If seeking proof then some finite-sample argument like Berry-Esseen theorem is needed to quantify the probability of the average is not Gaussian). And what is the mini-batch size used here? If it is too small then probably the error of taking the mini-batch SGD as Gaussian will be large.

(d) The importance/minimality of each individual bit number in C_o is not investigated, and the claim of the near minimality of C_o cannot hold according to the current experiments. The experiments of C_{+1} and C_{-1} do not exclude the possibility that some items (not the whole C_o) are minimal. More experiments (changing one or more items while fixing the others) are needed to show every item in C_o is minimal (or not). And which one of them is the most important for training/testing (how sensitive the training/testing performance is to each bit number)? Also in C_{-1} and C_{+1}, are target \beta, \eta changed? What are these changed values?

(e) From the Claim 1, it seems that these bit numbers are sufficient, and smaller than necessary to get the same training/testing performance (e.g., from the proof of Lemma 3, with the result \eta = 0.4% < 1%). So one question is what kind of \beta and \eta target are necessary for preserving the performance and what corresponding bit numbers are necessary to achieve the necessary \beta and \eta values?

(f) The computational cost definition looks different with Eq (3) in Sakr et al. 2017. Why?

Some typos:
1.Figure 2(a), B_{A_l} is 8 for Layer 1, but 9 in Appendix Table.
2. In the last third paragraph of Page 7, is it 2.6 = (148/56.5) instead of 2.6 * (148/56.5)? Same for the following numbers.

========================================
Revision: I have read the reply from the authors and it clarified several matters. I adjusted my rating of this paper (from 6 to 7).

---

> ### Author Response · Authors · 2018-11-11
> **Reply to AnonReviewer4 (Part 1/2)**
>
> Dear AnonReviewer4,
>
> Thank you very much for your detailed review! As we prepare our revised manuscript, we will address your questions and concerns. In this reply, we wish to provide a first response:
>
>
> - Reply to question (a)
>
> -- Thank you for this question. The motivation behind all the criteria is presented at the start of Section 3. These criteria are used to obtain a quantitative correspondence between the convergence behavior and the different precision assignments.
>
> As you have correctly pointed out, Criterion 1 (EFQN) is used to ensure that all feedforward quantization noise sources contribute equally to the p_m budget.
>
> Because FX numbers require a constant PDR, clipping of gradients is needed since their dynamic range is arbitrary. Ideally, a very small PDR would be preferred in order to obtain quantization steps of small magnitude, and hence less quantization noise. If you are familiar with signal processing theory, it is known that for a given quantizer, the signal-to-quantization-noise ratio (SQNR) is equal to SQNR (dB) = 6B + 4.78 – PAR where PAR is the peak-to-average ratio, proportional to the PDR. Thus, we would like to reduce the PDR as much as possible in order to increase the SQNR for a given precision. However, this comes at the risk of overflows (due to clipping). To address this trade-off between quantization noise and overflow errors, we introduce Criterion 2 (GC). The clipping rate \beta is used to quantify the impact of overflows for a given choice of PDR. To answer your question about the order of clipping, we are using simple max-clipping. That is, any scalar larger than the PDR in magnitude is clipped, while other scalars are unchanged.
>
> Since the back-propagation procedure is an iterative one, it is important to ensure that quantization bias does not accumulate in a positive feedback fashion. This is why we have introduced Criterion 3 (RQB) and \eta is used to quantify this bias as a function of quantization step size.
>
> Criterion 4 (BQN) is in fact an extension of Criterion 1 (EFQN), but for the back-propagation phase. Indeed, once the precision (and hence quantization noise) of weight gradients is set as per Criterion 3 (RQB), it is needed to ensure that the quantization noise source at the activation gradients would not contribute more noise to the updates. This criterion sets the quantization step of the activation gradients.
>
> Criterion 5 (AS) ties together feedforward and gradient precisions through the weight accumulators. It is required to increment/decrement the feedforward weights whenever the accumulated updates crossover the weight quantization threshold. This is used to set the PDR of the weight accumulators. Furthermore, since the precision of weight gradients has already been designed to account for quantization noise (through Criteria 2-4), we ensure that the accumulators do not cause additional noise. To do so the quantization step of the accumulator should be at most that of the weight gradient multiplied by the learning rate.
>
> As per your request, we will further motivate and include a summary of the above explanations in our revision.

---

> > ### Author Response · Authors · 2018-11-11
> > **Reply to AnonReviewer4 (Part 2/2)**
> >
> > - Reply to question (b)
> >
> > -- Good question. As described in Section 2.1 and as mentioned in the practical considerations paragraph following Claim 1, we use the following formula based on definition of precision, PDR, and LSB: B = log_2 (r/Delta) +1. We will make this mention more explicit following Claim 1.
> >
> >
> > - Reply to question (c)
> >
> > --Very good point. The CLT is used as approximation here, because we are averaging over the minibatch dimension. Fortunately, it is typical to use quite a large batch size for most deep learning applications. For instance, in our case we used a batch size of 256 throughout. Thus, by the Berry-Essen theorem, we expect the Kolmogorov-Smirnov distance between the true PDF and the Gaussian one to be small (as it decreases with a rate of n^-0.5). Using the Berry-Essen theorem, we will explicitly compute a bound on the approximation error and include in our revision. Thanks for this suggestion!
> >
> >
> > - Reply to question (d)
> >
> > -- That’s a very good point. We use the term ‘near’ minimal because, as you have correctly point out, simply comparing to C_{-1} and C_{+1} does not take into account all possible scenarios in between. However, please note that it is computationally prohibitive to perform a search over all precision assignments possible, since their number is exponentially increases with L (number of layers).
> >
> > We propose doing the following: we will randomly increment or decrement a random fraction of the precisions in C_o and repeat the experiments. If we can try this for several values of the fraction of precisions to alter, hopefully we can provide more evidence as to the ‘near’ minimality of C_{o}. These new experiment will be don during this rebuttal period, and we will be providing you with the new results.
> >
> > Also, to answer your question, target \beta is not altered because the PDRs are unchanged as mentioned in Section 4.2. Target \eta becomes 2.12% for C_{-1} as \Delta is multiplied by 2, and 0.16% for C_{+1} and \Delta is divided by 2 (these two numbers are obtained by evaluating \mu on page 13 for the new values of \Delta).
> >
> >
> > - Reply to question (e)
> >
> > -- Again, a very good question. In order to preserve performance, target values of \beta and \eta should be small. We do not provide an exact formula to relate these two quantities to the performance. Instead, we chose \beta = 5% and \eta = 1% for very low clipping rate and quantization bias, respectively. In practice, this choice of \beta and \eta was found to be adequate (as illustrated by our experiments).
> >
> > Further, the precisions chosen do achieve these values. As we have described in Section 2.1, FX numbers, by definition, have PDRs and LSBs equal to specific powers of 2. Should we modify the precision assignments to decrease the precision, then Claim 1 would no longer hold. For instance, as we mention in the response to question (d), decrementing the weight gradient precision cause \eta to become equal to 2.12%.
> >
> >
> > - Reply to question (f)
> >
> > -- Sakr et al. (2017) were only concerned with inference (feedforward) precision and did not consider the quantization of the back-propagation algorithm itself. In their setup, there is one set of dot products, those needed for the forward propagation. In our case, we have two other sets of dot products, those needed for the back-propagation and gradient computation. This is one difference between our works and theirs. Further, we have chosen to focus on the cost of multiplications (i.e., we have not included a term for the summations). The reason being representation quantization directly impacts the cost of multiplication and there are hardware/algorithmic ways to handle the cost of summations, which we leave as beyond the scope of our paper.
> >
> >
> > Finally, thanks for catching these typos, we will correct them in our revision.

---

> > > ### Author Response · Authors · 2018-11-22
> > > **Revision Uploaded**
> > >
> > > Dear AnonReviewer4,
> > >
> > > This is a note to let you know that we have uploaded our revision. To make it easy for you to track changes, we have typed all modifications with respect to the original manuscript in the blue color. In particular, we would like to mention:
> > >
> > > - We have added a new section in the Appendix (Page 11) to further motivate the five quantization criteria we used in our paper.
> > >
> > > - We have added a finite sample argument based on the Berry-Essen theorem in order to justify the use of the CLT in the proofs of Lemmas 2&3. This discussion can be found after the proofs (Page 16). We collected the necessary statistics from our baselines and evaluated the error bound on the CDFs and found a worst case approximation error of <0.09.
> > >
> > > - As per your suggestion, we have further investigated the minimality of the precision configuration. We have considered both random fractional precision perturbations and selective precision perturbations per tensor type. Additional experimental results are included in Appendix E (Page 23) in support of our claim of ‘near minimality’ of our precision assignment. One interesting finding from these new experiments is that weight and activation gradient precisions appear to be most sensitive as far as accuracy is concerned. We do hope this answers the original question in your review.
> > >
> > > We once again wish to thank you for your very nice review and comments which helped us enhance our work!

---

### Meta-Review · Area_Chair1 · 2018-12-17
**Contribution to the understanding of reduced precision training for deep networks**

**Confidence:** 4
**Recommendation:** Accept (Poster)

**Metareview:**

The paper investigates a detailed analysis of reduced precision training for a feedforward network, that accounts for both the forward and backward passes in detail.  It is shown that precision can be greatly reduced throughout the network computations while largely preserving training quality.  The analysis is thorough and carefully executed.

The technical presentation, including the motivation for some of the specific choices should be made clearer.  Also, the requirement that the network first be trained to convergence at full 32 bit precision is a significant limitation of the proposed approach (a weakness that is shared with other work in this area).  It would be highly desirable to find ways to bypass or at least mitigate this requirement, which would provide a real breakthrough rather than merely a solid improvement over competing work.

The reviewer disagreement revolves primarily around the clarity of the main technical exposition: there appears to be consensus that the paper is sound and provides a serious contribution to this area.

Although the persistent reviewer disagreement left this paper rated at the borderline, I am recommending acceptance, with the understanding that the authors will not disregard the dissenting review and strive to further improve the clarity of the presentation.